# NEURAL NETWORK EXPRESSIVE POWER ANALYSIS VIA MANIFOLD TOPOLOGY

## ABSTRACT

A prevalent assumption regarding real-world data is that it lies on or close to a lower-dimensional manifold. When deploying a neural network on data manifolds, the required size, i.e. the number of neurons, of the network heavily depends on the intricacy of the underlying latent manifold. While significant advancements have been made in understanding the geometric attributes of manifolds, it's essential to recognize that topology, too, is a fundamental characteristic of manifolds. In this study, we delve into a challenge where a classifier is trained with data on low-dimensional manifolds. We present an upper bound on the size of ReLU neural networks. This bound integrates the topological facets of manifolds, with empirical evidence confirming the tightness.

## 1 INTRODUCTION

The expressive power of deep neural networks (DNNs) is believed to play a critical role in their astonishing performance. Despite a rapidly expanding literature, the theoretical understanding of such expressive power remains limited. The well-known *universal approximation theorems* (Hornik, 1989; Cybenko, 1989; Leshno et al., 1993; Hanin, 2017) guarantee that neural networks can approximate vast families of functions with an arbitrarily high accuracy. However, the theoretical upperbound of the size of such networks is rather pessimistic; it is exponential to the input space dimension. Indeed, these bounds tend to be loose, because the analyses are often oblivious to the intrinsic structure of the data. Real-world data such as images are believed to live in a manifold of a much lower dimension (Roweis & Saul, 2000; van der Maaten & Hinton, 2008; Jolliffe & Cadima, 2016). Better bounds of network size can be achieved leveraging such manifold's structure. It has been shown that the network size can be bounded by exponential of the manifold's intrinsic dimension rather than the encompassing input space dimension (Chen et al., 2019; Schmidt-Hieber, 2019). However, the intrinsic dimension is only a small part of the manifold's property. It is natural to wonder whether other properties, such as topology and geometry of the manifold, may lead to improved bounds. Safran & Shamir (2016) demonstrate that to approximate the indicator function of a $d$-dimensional ball, one only needs a network of size quadratic to $d$. However, this work assumes a rather simplistic input. To extend to a more general setting, one needs to incorporate the topology and geometry of the manifold into the analysis.

Early research has probed the inherent structure of manifolds, with a particular emphasis on their geometric and topological characteristics. Notably, Federer (1959); Amenta & Bern (1998) introduce a pivotal curvature measure, known as the *reach*, which adeptly captures the geometric nuances of manifolds. This metric has been embraced in manifold learning studies (Narayanan & Niyogi, 2009; Narayanan & Mitter, 2010; Ramamurthy et al., 2019). On the topological front, descriptors like *Betti numbers* have been formalized in the language of algebraic topology to characterized the numbers of connected components and holes of a manifold (Hatcher, 2002; Bott et al., 1982; Munkres, 2018). In their seminal work, Niyogi et al. (2008) integrate manifold's geometry and topology, setting forth conditions for topologically faithful reconstructions grounded in geometric metrics. With the advent of the deep learning era, there has been a burgeoning interest in discerning the interplay between network size and manifold's intrinsic structural attributes. Beyond the previously mentioned studies on function approximation, where functions are characterized over manifolds, Dikkala et al. (2021); Schonsheck et al. (2019) investigate the interplay between network size and the geometric properties of manifolds in diverse deep learning applications.

Despite studies connecting network size and manifold geometry, there has not been any theoretical framework that successfully ties network size and topological traits together. When it comes to network size, the topological complexity of manifolds also plays a vital part in the learning problem. Empirical findings (Guss & Salakhutdinov, 2018; Naitzat et al., 2020) indicate that data with greater topological complexity necessitates a larger network. However, the theoretical relationship between topological attributes of data and network size remains an uncharted territory. This gap is likely due to the inherently discrete nature of topological descriptors, making their integration into neural network size analyses a challenging endeavor.

In response to these challenges, our research seeks to bridge this theoretical gap, presenting an innovative framework that integrates topology and neural network size. To capture the manifold's topological characteristics, we employ *Betti numbers*. These numbers specifically quantify the number of connected components and holes within the manifold. Our result reveals for the first time how the topology, as a global structural characterization of data manifold, affects the network expressiveness.

While topology offers global structural insights, it alone is not sufficient for bounding network size. To complement topology, we must also consider local properties, i.e., geometry. We introduce the condition number as a geometric measure, describing the manifold's overall flatness. Our theoretical analysis delivers an upper bound for network size controled by both Betti numbers (topology) and the condition number (geometry).

Our main theoretical result is informally summarized below.

**Main Theorem.** *(Informal) Let $\mathcal{M} = \mathcal{M}_1 \cup \mathcal{M}_2 \subset \mathbb{R}^D$ be a d-dimensional manifold ($d \leq D$, $d \leq 3$) with $\mathcal{M}_1$ and $\mathcal{M}_2$ representing two classes. Suppose $\mathcal{M}$ satisfies some mild manifold assumptions. For any $\delta > 0$, given a training set whose size is larger than a certain number $N(\delta)$, then for any $\epsilon > 0$, there exists a ReLU network classifier $g$ with depth at most $O(\log \beta + \log \frac{1}{\tau})$ and size at most $O(\frac{\beta^2}{\epsilon} + poly(\frac{1}{\tau}, \log \frac{1}{\delta}))$, such that $P(R(g) \leq \epsilon) > 1 - \delta$, where $R(g)$ is the true risk of g regarding any continuous distribution on $\mathcal{M}$. $\beta$ is the sum of Betti numbers and $\frac{1}{\tau}$ is the condition number.*

According to our bound, the network size scales quaratically in terms of total Betti number $\beta$. This can be validated by our empirical observations. Conversely, in terms of the condition number $\frac{1}{\tau}$, it scales as $O\left((\frac{1}{\tau})^{d^2/2}\right)$.

Our result provides a fresh theoretical perspective for further investigation of network expressiveness. In the future, the theory could potentially guide us towards designing more efficient neural networks based on manifold topology and geometry.

## 2 RELATED WORKS

**Network size with manifold geometry.** Multiple studies have formulated network size bounds across varied manifold learning contexts based on geometry. Schonsheck et al. (2019) establish a bound of $O(LdD\epsilon^{-d-d^2/2}(-\log^{1+d/2}\epsilon))$ on the network size for manifold reconstruction tasks. $L$ is the covering number in terms of the injectivity radius, a geometric property. They utilize an auto-encoder, denoted as $D \circ E$, for the reconstruction of a manifold. Both the encoder $E$ and the decoder $D$ are designed to function as homeomorphisms. As a result, the overarching objective is the construction of a homeomorphism within the same space, which elucidates the absence of topological considerations in their outcomes. Our findings include not only the homeomorphism but also the classification network, with the latter being influenced by the manifold's topology. Chen et al. (2019) demonstrate the existence of a network of size $O(\epsilon^{-d/n} \log \frac{1}{\epsilon} + D \log \frac{1}{\epsilon} + D \log D)$ that can approximate any smooth real function supported on a compact Riemannian manifold. In this context, $n$ denotes the order of smoothness of the function. Their primary objective is to illustrate that, in manifold learning, the manifold dimension chiefly determines network size, with only a marginal dependence on the ambient dimension. Moreover, their smoothness assumption is inapplicable to classification tasks, where the target function lacks continuity. Yet, the interplay between manifold properties and their impact on network size in manifold classification largely remains unexplored.

**Classifier learned on manifold.** Dikkala et al. (2021) investigates network size in classification contexts. However, their foundational assumption is that a manifold's essence can be distilled into just

two parameters: a centroid and a surrounding perturbation. They further assume there is a sensitive hashing property on manifolds. These assumptions are quite constrained, might not align with real-world complexities, and also overlooks the intrinsic properties of the manifold. Nevertheless, the aforementioned studies predominantly concentrate on network size and geometric traits, neglecting the equally critical role of topological features. Buchanan et al. (2021) established a lower bound on the size of classifiers for inputs situated on manifolds. However, their theoretical framework is restricted to smooth, regular simple curves, which exhibit uniform topology. This constraint negates the necessity to account for how variations in manifold topology might influence network size. Guss & Salakhutdinov (2018) provides empirical evidence showing that classifiers, when trained on data with higher Betti numbers, tend to have slower convergence rates. They also highlight that with rising topological complexity, smaller networks face challenges in effective learning. These findings underscore the need for a more comprehensive theoretical understanding.

There are some intriguing studies not primarily centered on manifold learning. Specifically, Bianchini & Scarselli (2014) establish a bound for the Betti number of a neural network's expression field based on its capacity. Nevertheless, their proposed bound is loose, and it exclusively addresses the regions a network can generate, neglecting any consideration of input manifold. Safran & Shamir (2016) explore the challenge of approximating the indicator function of a unit ball using a ReLU network. While their primary objective is to demonstrate that enhancing depth is more effective than expanding width, their approach has provided valuable insights. Naitzat et al. (2020) empirically examines the evolution of manifold topology as data traverses the layers of a proficiently trained neural network. We have adopted their concept of topological complexity. A number of studies, such as those by Hanin & Rolnick (2019) and Grigsby & Lindsey (2022), concentrate on exploring the potential expressivity of neural networks. However, these works primarily focus on the network's inherent capabilities without extensively considering the characteristics of the input data.

## 3 ASSUMPTIONS AND PRELIMINARIES

To explore the influence of manifold topology on network size, our analysis framework primarily leverages *manifold classification theorems*. Unlike the concept of classification as understood in machine learning, the classification referred to here pertains to theorems in topology. These theorems are focused on categorizing manifolds based on their structural properties. However, classifying manifolds beyond three dimensions is known to be equivalent to the word problem on groups (Markov, 1958), and is undecidable. The classification of 3-manifolds is also intricately complex (this is Thurston's geometrization, proved by Perelman (2003)), and beyond the scope of this work. As such, we restrict our study to *solid* manifolds of dimension three or less, which we define next.

**Assumption 1** (Solid Manifold). *In this paper, we assume the manifold $\mathcal{M}$ possesses a property as being 'solid'. The $d$-dimensional manifold $\mathcal{M} \subset \mathbb{R}^D$ is termed **solid** if the following conditions hold:*

*(A1): $\mathcal{M}$ can be embedded into $\mathbb{R}^d$, where $d \leq 3$ and $d \leq D$.*
*(A2): $\mathcal{M}$ is compact, orientable, and with boundary.*

**Remark 1.** *The solid manifold assumption aligns with our intuitive understanding of a 'solid shape', but embedded in a high-dimensional space. (A1) ensures that $\mathcal{M}$ possesses a $d$-dimensional volume in $\mathbb{R}^d$. This is essential for tasks like classification. This assumption rules out closed manifold like spheres. In certain cases, it is impossible to classify such manifolds because they lack a volume property. For instance, consider classifying a sphere along with its interior. Moreover, such perfectly closed shapes are seldom found in the real world due to sampling sparsity. (A2) imposes common topological properties on the manifold. This assumption excludes non-orientable manifolds such as the Klein bottle and projective planes.*

**Betti numbers.** We employ *Betti number* $\beta_k(\mathcal{M})$ to quantify topology of solid manifolds. $k$ is the dimension of that Betti number. 0-dimension Betti number $\beta_0(\mathcal{M})$ is the number of connected components in $\mathcal{M}$, and $\beta_k(\mathcal{M})$ ($k \geq 1$) can be informally described as the number of $k$-dimensional holes. 1-dimensional hole is a circle and 2-dimensional hole is a void. For the sake of coherence, we defer the formal definition of Betti numbers to Appendix A.1. Following Naitzat et al. (2020), we utilize the total Betti number of $\mathcal{M}$ as its topological complexity.

**Definition 1** (Topological Complexity). *$\mathcal{M}$ is a $d$-dimensional manifold. $\beta_k(\mathcal{M})$ is the $k$-dimensional Betti number of $\mathcal{M}$. The topological complexity is defined as*

$$\beta(\mathcal{M}) = \sum_{k=0}^{d-1} \beta_k(\mathcal{M}). \tag{1}$$

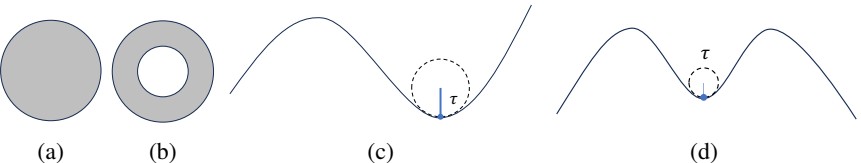

(a)      (b)         (c)               (d)

Figure 1: Illustration of Betti numbers and reach. (a) A 2-manifold with $\beta_0 = 1, \beta_1 = 0$. (b) A 2-manifold with $\beta_0 = 1, \beta_1 = 1$. (c) A 1-manifold with large reach. (d) A 1-manifold with small reach, which is the radius of the dashed circle.

**Reach and conditional number.** We then introduce metrics that encapsulate these geometric properties. For a compact manifold $\mathcal{M}$, the reach $\tau$ is the largest radius that the open normal bundle about $\mathcal{M}$ of radius $\tau$ is embedded in $\mathbb{R}^d$, i.e., no self-intersection.

**Definition 2** (Reach and Condition Number). *For a compact manifold $\mathcal{M} \subset \mathbb{R}^D$, let*

$$G = \{\mathbf{x} \in \mathbb{R}^D | \exists \mathbf{p}, \mathbf{q} \in \mathcal{M}, \mathbf{p} \neq \mathbf{q}, ||\mathbf{x} - \mathbf{p}|| = ||\mathbf{x} - \mathbf{q}|| = \inf_{\mathbf{y} \in \mathcal{M}} ||\mathbf{x} - \mathbf{y}||\}. \tag{2}$$

*The **reach** of $\mathcal{M}$ is defined as $\tau(\mathcal{M}) = \inf_{\mathbf{x} \in \mathcal{M}, \mathbf{y} \in G} ||\mathbf{x} - \mathbf{y}||$. The **condition number** $\frac{1}{\tau}$ is the inverse of the reach.*

Niyogi et al. (2008) prove that the condition number controls the curvature of the manifold at every point. A modest condition number $1/\tau$ signifies a well-conditioned manifold exhibiting low curvature.

**Problem setup.** In this paper, we examine the topology and geometry of manifolds in the classification setting. We have access to a training dataset $\{(\mathbf{x}_i, y_i) | \mathbf{x}_i \in \mathcal{M}, y_i \in [L]\}_{i=1}^n$, where $\mathcal{M} = \bigcup_{l=1}^{L} \mathcal{M}_l$. Each sample is drawn i.i.d. from a mixture distribution $\mu$ over $L$ disjoint manifolds with the corresponding label. For the simplification of notation, we build our theory on binary classification. It can be extended to multi-class without efforts in a one-verses-all setting. In binary case, the dataset is $\{(\mathbf{x}_i, y_i) | \mathbf{x}_i \in \mathcal{M}, y_i \in \{0, 1\}\}_{i=1}^n$, where $\mathcal{M} = \mathcal{M}_1 \cup \mathcal{M}_0 \in \mathbb{R}^D$. $\mathcal{M}_1$ and $\mathcal{M}_0$ are two disjoint $d$-dimensional solid manifolds representing two classes. Note that $\mathcal{M}$ is also a solid manifold (with two parts). The label $y_i$ is determined by the indicator function

$$I_{\mathcal{M}_1}(\mathbf{x}) = \begin{cases} 1, & \mathbf{x} \in \mathcal{M}_1, \\ 0, & \text{otherwise.} \end{cases} \tag{3}$$

A neural network $h(\mathbf{x}) : \mathbb{R}^D \to [0, 1]$ approaches the classification problem by approximating the indicator function $I_{\mathcal{M}_1}(\mathbf{x})$. In the scope of this study, we focus on neural networks utilizing the ReLU (Rectified Linear Unit) activation function.

**Definition 3** (Adapted from Arora et al. (2018)). *A ReLU multi-layer feed-forward network $h : \mathbb{R}^{w_0} \to \mathbb{R}^{w_{k+1}}$ with $k + 1$ layers is defined as*

$$h(\mathbf{x}) = h_{k+1} \circ h_k \circ \cdots \circ h_1(\mathbf{x}), \tag{4}$$

*where $h_i : \mathbb{R}^{w_{i-1}} \to \mathbb{R}^{w_i}, h_i(\mathbf{x}) = \sigma(W_i \mathbf{x} + b_i)$ for $1 \leq i \leq k$ are ReLU layers, and $h_{k+1} : \mathbb{R}^{w_k} \to \mathbb{R}^{w_{k+1}}, h_{k+1}(\mathbf{x}) = W_{k+1} \mathbf{x} + b_{k+1}$ is a linear layer. $\sigma$ is the ReLU activation function. The **depth** of the ReLU network is defined as $k + 1$. The **width** of the ReLU network is $\max\{w_1, \ldots, w_k\}$. The **size** of the ReLU network is $\sum_{i=1}^{k} w_i$.*

The approximation error of a ReLU network is determined by the true risk.

**Definition 4** (Approximation Error). *Let's consider the indicator function $I_{\mathcal{M}_1}$ for a manifold $\mathcal{M}_1$ in a binary classification problem where $\mathcal{M} = \mathcal{M}_1 \cup \mathcal{M}_0$. A neural network operates as a function $h(\mathbf{x}) : \mathcal{M} \to \mathbb{R}$. The approximation error of the neural network $h$ is then defined as:*

$$\text{True Risk: } R(h) = \int_{\mathcal{M}} (h - I_{\mathcal{M}_1})^2 \mu(\mathbf{x}) d\mathbf{x}. \tag{5}$$

*$\mu$ is any continuous distribution over $\mathcal{M}$.*

## 4 MAIN RESULTS

In this section, we explore how the topology of manifolds influence network size in classification scenarios. Our results, derived methodically through construction, follow two steps. First, we approximate a homeomorphism between the input manifold and a latent one; second, we carry out classification within this latent manifold. This latent manifold is designed to have simple geometric features, akin to those found in spheres and tori, while retaining the intrinsic topological characteristics of the original manifold. By design, the first phase is purely geometric, as the topological traits remain unaltered, while the subsequent classification phase is predominantly topological. Consequently, the required network size can be delineated into two distinct parts. We employ *Betti numbers* and the *condition number* as metrics to gauge topological and geometric complexities, respectively. Specifically, Betti numbers quantify the number of connected components and holes within the manifold, whereas the condition number characterizes the manifold's overall curvature.

### 4.1 COMPLEXITY ARISING FROM TOPOLOGY

In order to discard the geometric properties of the manifold, we concentrate on the basic shapes characterizing each dimension. These are balls in $\mathbb{R}^2$, $\mathbb{R}^3$ and solid torus in $\mathbb{R}^3$. We formally define them in Definition 5 and 6.

**Definition 5** (2-Dimensional Fundamental Manifold). *The 2-dimensional fundamental manifold is the disk $B_r^2(\mathbf{c}) = \{\mathbf{x} \in \mathbb{R}^2 : ||\mathbf{x} - \mathbf{c}||_2^2 \leq r\}$, with r the radius and $\mathbf{c}$ the center.*

**Definition 6** (3-Dimensional Fundamental Manifold). *The 3-dimensional fundamental manifolds are the following two classes,*

1. *A ball with radius r and center $\mathbf{c}$, $B_r^3(\mathbf{c}) = \{\mathbf{x} \in \mathbb{R}^3 : ||\mathbf{x} - \mathbf{c}||_2^2 \leq r\}$,*

2. *A solid torus (genus-1) with tunnel radius r and tunnel center radius R, $T_1 = \{P\mathbf{x} + \mathbf{c} : \mathbf{x} \in \mathbb{R}^3, x_3^2 + (\sqrt{x_1^2 + x_2^2} - R)^2 \leq r^2\}$, where P is the 3D rotation matrix.*

We refer to these manifolds as 'fundamental' because they encompass all potential topological configurations and present only trivial geometric features. This is evident when considering Betti numbers. For 2-dimensional solid manifolds, $\beta_0 = 1$ corresponds to a single $B_r^2$, while $\beta_1 = 1$ is depicted by excluding a small disk from a larger one. For 3-dimensional solid manifolds, representations for $\beta_0 = 1$ and $\beta_2 = 1$ are analogous. However, $\beta_1 = 1$ is typified by a solid torus. Put simply, solid manifolds can be characterized by their Betti numbers. Central to our approach is the application of the *manifold classification theorems* (Lee, 2010). Combined with our solid manifold assumption, we can prove that solid manifolds are homeomorphic to a combination of fundamental manifolds. We provide a formal exposition of this in Lemma 1, whose proof can be found in the Appendix A.4.

**Lemma 1** (Topological Representative). *If $\mathcal{M} \subset \mathbb{R}^D$ is a solid manifold, then there exists a manifold $\mathcal{M}' \subset \mathbb{R}^d (d \leq 3)$ that is homeomorphic to $\mathcal{M}$, where $\mathcal{M}'$ is constructed by a finite set of fundamental manifolds $\{\mathcal{F}_i\}_{i=1}^m$ via union and set subtraction. $m \leq \beta(\mathcal{M})$ is a constant integer. We term $\mathcal{M}'$ as the **topological representative** of $\mathcal{M}$.*

Our preliminary analysis focuses on the network size associated with these fundamental manifolds. Using this as a foundation, we then explore how various topological configurations of solid manifolds impact the size of neural networks. Given that both rotation and shift can be executed within a single linear layer, it suffices to examine a ball centered at the origin and a torus centered at the origin and aligned with the $z$-axis. Proposition 1 determines the network size required to approximate a $\mathbb{R}^d$ ball. While the original result is found in Safran & Shamir (2016), our study utilizes fewer parameters and offers a different way to approximate the threshold function. Proposition 2 outlines a network size bound for the approximation of a solid torus.

**Proposition 1** (Approximating a $\mathbb{R}^d$ Ball, adapted from Theorem 2 in Safran & Shamir (2016)). *Given $\epsilon > 0$, there exists a ReLU network $h : \mathbb{R}^d \to \mathbb{R}$ with 3 layers and with size at most $4d^2r^2/\epsilon + 2d + 2$, which can approximate the indicator function $I_{B_r^d}$ within error $R(h) \leq \epsilon$ for any continuous distribution $\mu(\mathbf{x})$.*

**Proposition 2** (Approximating a Solid Torus). *Given $\epsilon > 0$, there exists a ReLU network $h : \mathbb{R}^3 \to \mathbb{R}$ with 5 layers and with size at most $\frac{2d}{\epsilon}(4(d-1)(R+r)^2 + 8r^2 + \frac{r}{\sqrt{R-r}}) + 9$, which can approximate the* indicator function *$I_{T_1}$ within error $R(h) \leq \epsilon$ for any continuous distribution $\mu(\mathbf{x})$.*

Proposition 1 and 2 address the network size associated with approximating fundamental manifolds in $\mathbb{R}^2$ and $\mathbb{R}^3$. Detailed proofs can be found in the Appendix A.3. Building on this, we can infer the network size for approximating topological representatives by combining the complexities of approximating fundamental manifolds with those of union and set subtraction operations. This consolidated insight is captured in Theorem 1.

**Theorem 1** (Complexity Arising from Topology). *Suppose $\mathcal{M}'$ is the topological representative of a solid manifold. Given $\epsilon > 0$, there exists a ReLU network $h : \mathbb{R}^d \to \mathbb{R}$ ($d \leq 3$) with depth at most $O(\log \beta)$ and size at most $O(\frac{d^2 \beta^2}{\epsilon})$, that can approximate the* indicator function *$I_{\mathcal{M}'}$ with error $R(h) \leq \epsilon$ for any continuous distribution $\mu$ over $\mathbb{R}^d$. $\beta$ is the topological complexity of $\mathcal{M}'$.*

The detailed proof can be found in Appendix A.4. This theorem offers an upper bound on the network size required to approximate the indicator function of a topological representative. It is important to note that this captures the full range of complexities arising from the topology of a solid manifold $\mathcal{M}$, given that $\mathcal{M}$ and $\mathcal{M}'$ are homeomorphic. To the best of our knowledge, this is the first result bounding neural network size in terms of a manifold's Betti numbers.

## 4.2 OVERALL COMPLEXITY

General solid manifolds, due to their inherent complexity, often defy explicit expression. This hinders the direct use of function analysis for approximating their indicator functions, as was done in previous studies. To tackle this issue, we construct a homeomorphism, a continuous two-way transformation, between the solid manifold and its corresponding topological representative. This method only alters the geometric properties, preserving the object's topological attributes. Therefore, the network size in approximating the indicator function of general solid manifolds are constructed by the approximation of the homeomorphism and the classification of topological representatives. The network size of constructing the homeomorphism is exclusively influenced by the geometric properties, whereas the network size of classifying topological representatives pertains solely to topological properties. The latter we already figured in previous section. This methodology enables us to distinguish the influence of topology and geometry of manifold on classifiers. In this section, we aim to obtain the overall network size for a classifier.

To build a homeomorphism from $\mathcal{M}$, we first need to recover the homology of $\mathcal{M}$. The subsequent proposition outlines a lower limit for the number of points essential to recover the homology of the initial manifold $\mathcal{M}$.

**Proposition 3** (Theorem 3.1 in Niyogi et al. (2008)). *Let $\mathcal{M}$ be a compact submanifold of $\mathbb{R}^D$ with condition number $1/\tau$. Let $X = \{\mathbf{x}_1, \mathbf{x}_2, ..\mathbf{x}_n\}$ be a set of $n$ points drawn in i.i.d. fashion according to the uniform probability measure on $\mathcal{M}$. Let $0 < \epsilon < \frac{\tau}{2}$. Let $U = \bigcup_{\mathbf{x} \in X} B_\epsilon(\mathbf{x})$ be a corresponding random open subset of $\mathbb{R}^D$. Then for all*

$$n > \lambda_1(log(\lambda_2) + log(\frac{1}{\delta})), \tag{6}$$

*$U$ is a $\epsilon$-cover of $\mathcal{M}$, and the homology of $U$ equals the homology of $\mathcal{M}$ with high confidence (probability $> 1 - \delta$). Here*

$$\lambda_1 = \frac{vol(\mathcal{M})}{(cos^d\theta_1)vol(B_{\epsilon/4}^d)} \ and \ \lambda_2 = \frac{vol(\mathcal{M})}{(cos^d\theta_2)vol(B_{\epsilon/8}^d)}, \tag{7}$$

*$\theta_1 = arcsin(\epsilon/8\tau)$ and $\theta_2 = arcsin(\epsilon/16\tau)$. $d$ is the latent dimension of $\mathcal{M}$, and $vol(B_\epsilon^d)$ denotes the $d$-dimensional volume of the standard $d$-dimensional ball of radius $d$. $vol(\mathcal{M})$ is the $d$-dimensional volume of $\mathcal{M}$.*

This result stipulates a lower bound for the training set size necessary to recover the homology of the manifold, which is the foundation to learn the homeomorphism between a solid manifold $\mathcal{M}$ and its topological representative $\mathcal{M}'$. However, directly constructing this homeomorphism remains

challenging. As a workaround, we develop a simplicial homeomorphism to approximate the genuine homeomorphism. Notably, this simplicial approach lends itself readily to representation via neural networks.

Combined with the topological representative classification network in Theorem 1, we can construct a classification network for solid manifolds in the following manor, as depicted in Figure 2. Initially, we project $\mathcal{M}$ to its simplicial approximation $|K|$ using a neural network $N_p$. This is succeeded by a network $N_\phi$ that facilitates the simplicial homeomorphism between $|K|$ and $|L|$, the latter being the simplicial approximation of the topological representative $\mathcal{M}'$. Finally, a network $h$ is utilized to classify between $|L_1|$ and $|L_0|$. Consequently, the network's size is divided into two main parts: one focused on complexities related to geometric attributes and the other concerning topological aspects. This distinction separates topology from geometry in classification problems.

In Theorem 2, we design such a neural network based on this training set, ensuring that approximation errors are effectively controlled. The detailed proof is provided in Appendix A.5. Our proof strategy begins with the construction of a ReLU network, followed by an evaluation of the network's size. Subsequently, we place bounds on the involved approximation errors.

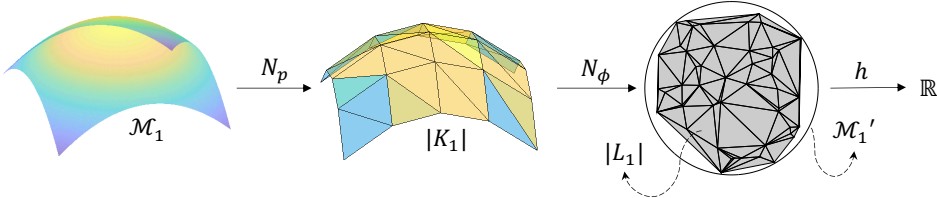

Figure 2: Construction of the network $g$. While the diagram illustrates only the process for the manifold of the positive class, the procedure for $\mathcal{M}_0$ mirrors this operation identically.

**Theorem 2** (Main Theorem). *Let $\mathcal{M} = \mathcal{M}_1 \bigcup \mathcal{M}_0 \subset \mathbb{R}^D$ be a d-dimensional solid manifold. $\mathcal{M}_1$ and $\mathcal{M}_0$ are two disjoint sub-manifolds of $\mathcal{M}$ representing two classes and both are also solid. The condition number of $\mathcal{M}$ is $\frac{1}{\tau}$ and the total Betti number of $\mathcal{M}_1$ is $\beta$. Given a training set $\{(\mathbf{x}_i, y_i) | \mathbf{x}_i \in \mathcal{M}, y_i \in \{0, 1\}\}_{i=1}^n$, where $\mathbf{x}_i$ are sampled i.i.d. from $\mathcal{M}$ by a uniform distribution, and $y_i = I_{\mathcal{M}_1}(\mathbf{x}_i)$. For any $\delta > 0$, if inequality (6) holds, then for any $\epsilon > 0$, there exists a ReLU network $g$ with depth at most $O(\log \beta + d \log \frac{1}{\tau} + \log \log \frac{1}{\tau\delta})$ and size at most $O(\frac{d^2\beta^2}{\epsilon} + \tau^{-d^2/2} \log^{d/2} \frac{1}{\tau\delta} + D\tau^{-d} \log \frac{1}{\tau\delta})$, such that*

$$P(R(g) \leq \epsilon) > 1 - \delta, \tag{8}$$

*where $R(g) = \int_{\mathcal{M}} (g - I_{\mathcal{M}_1})^2 \mu(\mathbf{x}) d\mathbf{x}$ with any continuous distribution $\mu$.*

Upon examining the depth and size of the neural network, it becomes evident that the topological complexity, denoted by $\beta$, and the geometric complexity, symbolized by $\tau$, are distinctly delineated. The topological complexity contributes $O\left(\frac{d^2\beta^2}{\epsilon}\right)$ to the overall network size. In contrast, geometry contributes $O\left(\tau^{-d^2/2} \log^{d/2} \frac{1}{\tau\delta} + D\tau^{-d} \log \frac{1}{\tau\delta}\right)$.

It is important to note that our result is constructed as an upper bound. In practical scenarios, a neural network trained without specific constraints might not follow a strict sequence of first learning a homeomorphism to latent representations and then executing classification. Instead, it could adopt a more integrated approach, intertwining classification information during the representation learning process. This means that the actual network size could be significantly less than our provided bound. However, the tightness of topological bound $O\left(\frac{d^2\beta^2}{\epsilon}\right)$ can be empirically verified with fixed dimension. We delve into this in the subsequent section.

## 5  EMPIRICAL VALIDATION

In this section, we present numerical results that showcase our topological bound $O\left(\frac{d^2\beta^2}{\epsilon}\right)$ in fixed dimension. Even though this bound is derived through construction and serves as an upper limit, it is

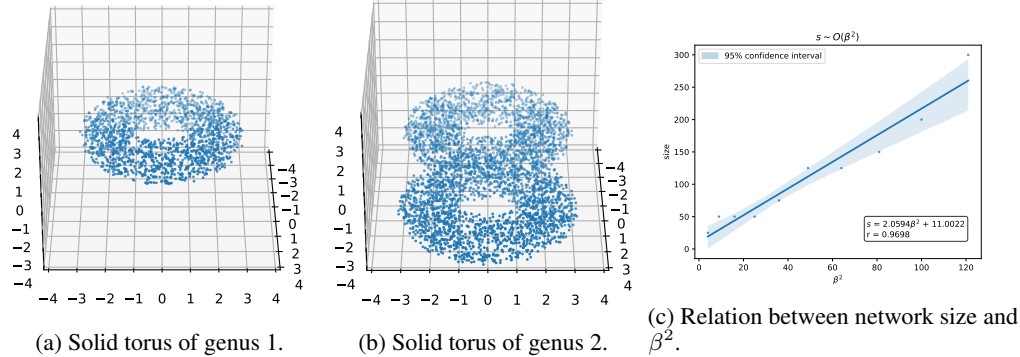

(a) Solid torus of genus 1.  (b) Solid torus of genus 2. (c) Relation between network size and $\beta^2$.

Figure 3: Validation of topological complexity. (a,b) are samples from two solid tori with different topological complexity. Samples from the negative class, which is not shown, are sampled uniformly from the background. (c) showcases the linear regression results between the network size and the square of topological complexity. A 5-layer neural network with adaptive width is trained to fit tori of varying genus $g$. The width of the network is increased until the training accuracy exceeds 0.95.

intriguing to discover that the bound is tight and can be readily observed in experimental settings. In practical scenarios, when training networks of varying sizes on data drawn from manifolds with different total Betti numbers $\beta$, and achieving the same error rate, we anticipate a linear relationship between the network size and $\beta^2$.

We utilize manifolds characterized as solid genus-$g$ tori, with $g$ spanning from 1 to 10. Each genus-$g$ torus is synthesized by overlapping two identical tori. For each torus, we consistently sample $g \times 10^4$ points from a surrounding bounding box. The labels for these points are generated using the indicator function of the solid torus.

For training, we deploy a 5-layer ReLU network, gradually increasing its width until the training accuracy surpasses 0.95. Figure 3c presents a regression line charting the relationship between network size and the squared topological complexity, $\beta^2$. This regression underscores a pronounced linear association between network size $s$ and $\beta^2$, with a correlation coefficient $r = 0.9698$.

## 6 CONCLUSION

In this study, we delved into the intricate relationship between network size, and both geometric and topological characteristics of manifolds. Our findings underscored that while many existing studies have been focused on geometric intricacies, it is important to also appreciate the manifold's topological characteristics. These characteristics not only offer an alternative perspective on data structures but also influence network size in significant ways.

Although our proposed network size bounds represent theoretical upper limits, our empirical validations concerning the topological bound are promising. Nevertheless, real-world implementations may yield efficiencies beyond these confines. To attain a more direct and refined theoretical bound, we may need more comprehensive descriptors of manifolds that go beyond merely the Betti numbers and the condition number. We leave this exploration for future work. We hope that our study acts as a catalyst for further research, pushing the boundaries of manifold learning and its applications in modern AI systems.

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

# A APPENDIX

In this section, we formally prove the theoretical findings presented in the primary manuscript. Initially, we utilize some necessary definitions and existing results. Then we prove the network size bound for fundamental solid manifolds and general topological representatives.

## A.1 ADDITIONAL DEFINITIONS

The first definition is *Betti number*, which is a vital part of this paper. The $k$-th Betti number is defined as the rank of $k$-th homology group. Therefore, we have to properly define homology group first. Our definition follows Hatcher (2002) but is tailed for simplification. We first define *simplicial homology* for simplicial complexes. (Actually for $\Delta$-complexes. we simplify it to avoid introducing $\Delta$-complexes.) Then extend it to *singular homology* that can be applied to manifolds.

**Simplicial Homology.** Let $K$ be a simplicial complex, and let $K^k$ be the set of all $k$-dimensional simplices in $K$. The set of $K^k$ together with the field $\mathbb{Z}_2$ forms a group $C_k(K)$. It is a vector space defined on $\mathbb{Z}_2$ with $K^k$ as a basis. The element of $C_k(K)$ is called a $k$-chain. Let $\sigma \in K^k$ be a $k$-simplex. The boundary $\partial_k(\sigma)$ is the collection of its $(k-1)$-dimensional faces, which is a $k-1$-simplicial complex. The boudnary operator is linear, i.e.

$$\partial_k(z_1\sigma_1 + z_2\sigma_2) = z_1\partial_k(\sigma_1) + z_2\partial_k(\sigma_2).$$

The boundary operator $\partial_k : C_k(K) \to C_{k-1}(K)$ introduces a chain complex

$$\cdots \to C_d \xrightarrow{\partial_d} C_{d-1} \xrightarrow{\partial_{d-1}} C_{d-2} \to \cdots \to C_0 \xrightarrow{\partial_0} \emptyset.$$

$d$ is the maximum dimension of $K$. Ker $\partial_k$ is the collection of $k$-chains with empty boundary and Im $\partial_k$ is the collection of $(k-1)$-chains that are boundaries of $k$-chains. Then we can define the $k$-th homology group of the chain complex to be the quotient group $H_k = \text{Ker } \partial_k / \text{Im } \partial_{k+1}$. The $k$-th Betti number is defined by

$$\beta_k = \text{rank } H_k.$$

**Singular Homology.** Given a topological space $X$, the $k$-th singular chain group $C_k(X)$ is defined as the free Abelian group generated by the continuous maps $\phi : K^k \to X$, where $K^k$ is the standard $k$-simplex in $\mathbb{R}^k$. Each such map is referred to as a singular $k$-simplex in $X$.

A boundary operator $\partial_k : C_k(X) \to C_{k-1}(X)$ can be defined as:

$$\partial\phi = \sum_{i=0}^{n}(-1)^i\phi|_{[v_0,\cdots,\hat{v}_i,\cdots,v_n]},$$

where $\phi|_{[v_0,\cdots,\hat{v}_i,\cdots,v_n]}$ represents the restriction of $\sigma$ to the $i$-th face of $K^k$.

The $k$-th singular homology group $H_k(X)$ is then represented as the quotient:

$$H_k(X) = \text{Ker } \partial_k / \text{Im } \partial_{k+1}.$$

The $k$-th Betti number is still defined as $\beta_k = \text{rank } H_k(X)$.

## A.2 PRELIMINARY RESULTS

We present some pre-established results regarding the network size associated with learning a 1-dimensional piecewise linear function, as well as basic combinations of functions.

**Lemma 2** (Theorem 2.2. in Arora et al. (2018))**.** *Given any piecewise linear function $\mathbb{R} \to \mathbb{R}$ with $p$ pieces there exists a 2-layer ReLU network with at most $p$ nodes that can represent $f$. Moreover, if the rightmost or leftmost piece of a the piecewise linear function has $0$ slope, then we can compute such a $p$ piece function using a 2-layer ReLU network with size $p - 1$.*

**Lemma 3** (Function Composition, Lemma D.1. in Arora et al. (2018))**.** *If $f_1 : \mathbb{R}^d \to \mathbb{R}^m$ is represented by a ReLU DNN with depth $k_1 + 1$ and size $s_1$, and $f_2 : \mathbb{R}^m \to \mathbb{R}^n$ is represented by a ReLU DNN with depth $k_2 + 1$ and size $s_2$, then $f_2 \circ f_1$ can be represented by a ReLU DNN with depth $k_1 + k_2 + 1$ and size $s_1 + s_2$.*

**Lemma 4** (Function Addition, Lemma D.2. in Arora et al. (2018)). *If $f_1 : \mathbb{R}^n \to \mathbb{R}^m$ is represented by a ReLU DNN with depth $k + 1$ and size $s_1$, and $f_2 : \mathbb{R}^n \to \mathbb{R}^m$ is represented by a ReLU DNN with depth $k + 1$ and size $s_2$, then $f_1 + f_2$ can be represented by a ReLU DNN with depth $k + 1$ and size $s_1 + s_2$.*

**Lemma 5** (Taking maximums, Lemma D.3. in Arora et al. (2018)). *Let $f_1, ..., f_m : \mathbb{R}^n \to \mathbb{R}$ be the functions that each can be represented by ReLU networks with depth $k_i + 1$ and size $s_i$, $i = 1, ..., m$. Then the function $f : \mathbb{R}^n \to \mathbb{R}$ defined as $f = \max\{f_1, ..., f_m\}$ can be represented by a ReLU network of depth at most $\max\{k1, ..., k_m\} + \log(m) + 1$ and size at most $s_1 + ... + s_m + 4(2m - 1)$.*

We proceed to disclose the network size involved in learning a 1-dimensional Lipschitz function.

**Lemma 6** (Lipschitz Function Approximation, adapted from Lemma 11 in Eldan & Shamir (2016)). *For any $L$-Lipschitz function $f : \mathbb{R} \to \mathbb{R}$ which is constant outside a bounded interval $[a, b]$, and for any $\epsilon > 0$, there exits a two-layer ReLU network $h(x)$ with at most $\lceil L(b - a)/\epsilon \rceil + 1$ nodes, such that*

$$\sup_{x \in \mathbb{R}} |f(x) - h(x)| < \epsilon.$$

*Proof.* We follow the original proving idea but adapt it for better understanding. We prove the lemma by estimate the Lipschitz function by a piece-wise linear function within error $\epsilon$ and use a two-layer ReLU network to represent the piece-wise linear function.

We first cut the interval equally into $m$ sections $[a, b] = \bigcup_{i=1}^{m} [a + (i - 1)\delta, a + i\delta]$, where $\delta = (b - a)/m$. For each interval $I_i = [a + (i - 1)\delta, a + i\delta]$, we denote $f_i(x) = f|_{I_i}$. Then $\forall x_1, x_2 \in I_i$, $|f_i(x_1) - f_i(x_2)| \le L|x_1 - x_2| \le L\delta$. Let $h_i(x)$ be the linear function defined on this interval and connect $(a + (i - 1)\delta, f_i(a + (i - 1)\delta))$ and $(a + i\delta, f_i(a + i\delta))$. Then we can bound the difference between $f_i(x)$ and $h_i(x)$ by

$$
\begin{aligned}
|f_i(x) - h_i(x)| &\le \max\{|\max f_i(x) - \min h_i(x)|, |\min f_i(x) - \max h_i(x)|\} \\
&= \max\{|\max f_i(x) - f_i(a + (i - 1)\delta)|, |\min f_i(x) - f_i(a + i\delta))|\} \qquad \text{(A.1)} \\
&\le L\delta.
\end{aligned}
$$

The second line assumes $h_i(x)$ is non-decreasing. The other case can also be easily verified. By setting $m = \lceil \frac{L(b-a)}{\epsilon} \rceil$, for every interval, the error is controlled by $\epsilon$. Let $h(x)$ be the collection of all $h_i$ and also the constant outside of $[a, b]$, so we have $\sup_{x \in \mathbb{R}} |f(x) - h(x)| < \epsilon$.

$h(x)$ is a piece-wise linear function with $m + 2$ pieces. According to Lemma 2, there exists a 2-layer ReLU network with at most $m + 1$ pieces that can represent $h(x)$. Proof done. $\qquad \square$

## A.3 Approximating Fundamental Solid Manifolds

Now we are in a good position to prove Proposition 1 and 2.

**Proposition 1** (Approximating a $\mathbb{R}^d$ Ball, adapted from Theorem 2 in Safran & Shamir (2016)). *Given $\epsilon > 0$, there exists a ReLU network $h : \mathbb{R}^d \to \mathbb{R}$ with 3 layers and with size at most $4d^2r^2/\epsilon + 2d + 2$, which can approximate the indicator function $I_{B_r^d}$ within error $R(h) \le \epsilon$ for any continuous distribution $\mu(\mathbf{x})$.*

*Proof.* We generally follow the original proof but derive a slightly different bound with fewer parameters. The proof is organized by first using a non-linear layer to approximate a truncated square function and then using another non-linear layer to approximate a threshold function. Consider the truncated square function

$$l(x; r) = \min\{x^2, r^2\}. \qquad \text{(A.2)}$$

Clearly $l(x; r)$ is a Lipschitz function with Lipschitz constant $2r$. Applying Lemma 6, we have a 2-layer ReLU network $h_{11}$ that can approximate $l(x; r)$ with

$$\sup_{x \in \mathbb{R}} |h_{11}(x) - l(x)| \le \epsilon_1, \qquad \text{(A.3)}$$

with at most $2r^2/\epsilon_1 + 2$ nodes. Now for $\mathbf{x} \in \mathbb{R}^d$, let

$$h_1(\mathbf{x}) = \sum_{i=1}^{d} h_{1i}(x_i). \qquad \text{(A.4)}$$

Note that $h_1$ is also a 2-layer network because no extra non-linear operation is introduced in equation A.4, and has size at most $2dr^2/\epsilon_1 + 2d$. This can also be verified by Lemma 4. Let

$$L(\mathbf{x}) = \sum_i^d L(x_i; r), \tag{A.5}$$

and we have

$$\sup_{\mathbf{x}} |h_1(\mathbf{x}) - L(\mathbf{x})| \le d\epsilon_1. \tag{A.6}$$

Let $\epsilon_1 = d\epsilon_1$, then $h_1$ has size at most $2d^2r^2/\epsilon_1 + 2d$. Although $L(\mathbf{x})$ is different from $\sum x_i^2$, the trick here is to show $B_r^d = \{\mathbf{x} : L(\mathbf{x}) \le r^2\}$.

On the one hand, if $L(rvx) \le r^2$, remember that

$$L(\mathbf{x}) = \sum_{i=1}^d \min\{x_1^2, r^2\} \le r^2. \tag{A.7}$$

This means for all $x_i$, $x_i \le r^2$. Therefore, $L(\mathbf{x}) = \sum_{i=1}^d x_i^2$. On the other, $L(\mathbf{x}) > r^2$ only happens when there exists a $i$, such that $x_i^2 > r^2$. Thus, $\mathbf{x} \notin B_r^d$. Consequently, one can represent $I_{B_r^d}$ by $L(\mathbf{x}) \le r^2$.

The next step towards this proposition is to construct another 2-layer ReLU network to threshold $L(\mathbf{x})$. Consider

$$f(x) = \begin{cases} 1, & x < r^2 - \delta, \\ \frac{r^2-x}{\delta}, & x \in [r^2 - \delta, r^2], \\ 0, & x > r^2. \end{cases} \tag{A.8}$$

Note that $f$ is a 3-piece piece-wise linear function that approximates a threshold function. According to Lemma 2, a 2-layer ReLU network $h_2$ with size 2 can represent $f$. The function $f \circ L(\mathbf{x})$ can then be estimated by a 3-layer network $h = h_2 \circ h_1$, whose size is $2d^2r^2/\epsilon_1 + 2d + 2$. The next step is to bound the error between $h$ and $I_{B_r^d}$. We consider the $L_2$-type bound $||h(\mathbf{x}) - I_{B_r^d}(\mathbf{x})||_{L_2(\mu)} = \int_{\mathbb{R}^d} (h(\mathbf{x}) - I_{B_r^d}(\mathbf{x}))^2 \mu(\mathbf{x})d\mathbf{x}$. We divide the integral into two parts

$$\begin{aligned} &||h(\mathbf{x}) - I_{B_r^d}(\mathbf{x})||_{L_2(\mu)} \\ &\le ||f \circ L(\mathbf{x}) - I_{B_r^d}(\mathbf{x})||_{L_2(\mu)} + ||f \circ L(\mathbf{x}) - h_2 \circ h_1(\mathbf{x})||_{L_2(\mu)} \\ &= I_1 + I_2. \end{aligned} \tag{A.9}$$

Since $\mu(\mathbf{x})$ is continuous, there exists $\delta$ such that

$$\int_{S_\delta} \mu(\mathbf{x})d\mathbf{x} \le \epsilon_2. \tag{A.10}$$

$S_\delta = \{\mathbf{x} \in \mathbb{R}^3 : r^2 - \delta \le \sum_{i=1}^d x_i^2 \le r^2\}$. Combine equation A.8 we have

$$\begin{aligned} I_1 &= \int_{\mathbb{R}^3} (f \circ L(\mathbf{x}) - I_{B_r^d}(\mathbf{x}))^2 \mu(\mathbf{x})d\mathbf{x} \\ &= \int_{S_\delta} (f \circ L(\mathbf{x}) - I_{B_r^d}(\mathbf{x}))^2 \mu(\mathbf{x})d\mathbf{x} \\ &= \int_{S_\delta} (f \circ L(\mathbf{x}) - 1)^2 \mu(\mathbf{x})d\mathbf{x} \\ &\le \int_{S_\delta} \mu(\mathbf{x})d\mathbf{x} \\ &\le \epsilon_2. \end{aligned} \tag{A.11}$$

The first inequality is because $f \in [0, 1]$, such that $(f \circ L(\mathbf{x}) - 1)^2 \le 1$. The second part of the error can be easily bounded by its infinity norm.

$$I_2 = ||f \circ L(\mathbf{x}) - h_2 \circ h_1(\mathbf{x})||_{L_2(\mu)} \le ||f \circ L(\mathbf{x}) - h_2 \circ h_1(\mathbf{x})||_\infty \le \epsilon_1. \tag{A.12}$$

The last inequality is because $h_2$ is the exact representation of $f$, the error only occurs between $L(\mathbf{x})$ and $h_1$. Combine A.11 and A.12, and let $\epsilon_1 = \epsilon_2 = \epsilon/2$, we have

$$\|h(\mathbf{x}) - I(\mathbf{x})\|_{L_2(\mu)} \leq \epsilon. \tag{A.13}$$

The size of network $h$ is then bounded by $4d^2r^2/\epsilon + 2d + 2$. □

**Proposition 2** (Approximating a Solid Torus). *Given $\epsilon > 0$, there exists a ReLU network $h : \mathbb{R}^3 \to \mathbb{R}$ with 5 layers and with size at most $\frac{2d}{\epsilon}(4(d-1)(R+r)^2 + 8r^2 + \frac{r}{\sqrt{R-r}}) + 9$, which can approximate the [indicator function]{.blue} $I_{T_1}$ within error $R(h) \leq \epsilon$ for any continuous distribution $\mu(\mathbf{x})$.*

*Proof.* The proof is done by two steps. We first use layers to estimate a truncated function. Then estimate a threshold function by another layer.

Consider the truncated square function and root function,

$$l_1(x; \gamma) = \min\{x^2, \gamma^2\},$$
$$l_2(x; \gamma_1, \gamma_2) = \min\{\max\{\sqrt{x}, \gamma_1\}, \gamma_2\}, (\gamma_1 < \gamma_2).$$

The Lipschitz constants for $l_1$ and $l_2$ are $2\gamma$ and $\frac{1}{2\sqrt{\gamma_1}}$, respectively. By Lemma 6, there is a 2-layer ReLU network to approximate $l_1$ and $l_2$ with size $\lceil 4\gamma^2/\epsilon_1 \rceil + 1$ and $\lceil (\gamma_2 - \gamma_1)/(2\epsilon_1\sqrt{\gamma_1}) \rceil + 1$, respectively. Let

$$L(\mathbf{x}) = l_1(x_3; r) + l_1(l_2(l_1(x_1; R+r) + l_1(x_2; R+r); R-r, R+r) - R; r). \tag{A.14}$$

Then it is time to show $T_1 = \{\mathbf{x} \in \mathbb{R}^3 : x_3^2 + (\sqrt{x_1^2 + x_2^2} - R)^2 \leq r^2\} = \{\mathbf{x} : L(\mathbf{x}) \leq r^2\}$. For $\mathbf{x} \in I_{T_1}(\mathbf{x})$, the following inequalities hold

$$x_1^2 \leq (R+r)^2, x_2^2 \leq (R+r)^2, \tag{A.15}$$

$$R + r \geq \sqrt{x_1^2 + x_2^2} \geq R - r, \tag{A.16}$$

$$x_3^2 \leq r^2, (\sqrt{x_1^2 + x_2^2} - R)^2 \leq r^2. \tag{A.17}$$

These indicate that $L(\mathbf{x}) = x_3^2 + (\sqrt{x_1^2 + x_2^2} - R)^2 \leq r^2$, when $\mathbf{x} \in T_1$. And when $\mathbf{x} \notin T_1$, if $L(\mathbf{x}) = x_3^2 + (\sqrt{x_1^2 + x_2^2} - R)^2$ still holds, clearly $L(\mathbf{x}) > r^2$. Otherwise, one of the inequalities in A.15, A.16 and A.17 must break. If one of A.17 breaks, then clearly $L(\mathbf{x}) > r^2$. If A.16 does not hold, then $(\sqrt{x_1^2 + x_2^2} - R)^2 > r^2$, resulting $L(\mathbf{x}) > r^2$. The violation of A.15 resulting violation of A.16, which then leads to $L(\mathbf{x}) > r^2$.

To see how a ReLU network can estimate $L(\mathbf{x})$, we start by estimating each of its component. We define the following 2-layer networks. To make the overall network take $\mathbf{x} \in \mathbb{R}^3$ as input, we consider

| Network | Target | Size |
|---------|--------|------|
| $h_{11}$ | $l_1(x_1; R+r)$ | $s_{11} = \lceil 4(R+r)^2/\epsilon_1 \rceil + 1$ |
| $h_{12}$ | $l_1(x_2; R+r)$ | $s_{12} = \lceil 4(R+r)^2/\epsilon_1 \rceil + 1$ |
| $h_2$ | $l_2(x_1; R-r, R+r)$ | $s_2 = \lceil r/(\epsilon_1\sqrt{R-r}) \rceil + 1$ |
| $h_{31}$ | $l_1(x_1; , r)$ | $s_{31} = \lceil 4r^2/\epsilon_1 \rceil + 1$ |
| $h_{32}$ | $l_1(x_3; , r)$ | $s_{32} = \lceil 4r^2/\epsilon_1 \rceil + 1$ |

the following structure,

By Lemma 4 and Lemma 3 and the given structure, a ReLU network $\tilde{L}$ with depth 4 and size $(d-1)s_{11} + s_2 + s_{31} + s_{32} + 2$, where $d = 3$, can approximate $L(\mathbf{x})$ such that

$$\sup_{\mathbf{x}} |L(\mathbf{x}) - \tilde{L}(\mathbf{x})| \leq d\epsilon_1. \tag{A.18}$$

The next step is to threshold $L(\mathbf{x})$. Consider a function

$$f(x) = \begin{cases} 1, & x < r^2 - \delta, \\ \frac{r^2 - x}{\delta}, & x \in [r^2 - \delta, r^2], \\ 0, & x > r^2. \end{cases} \tag{A.19}$$

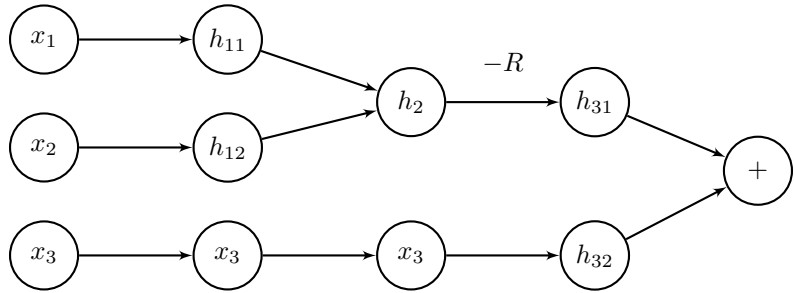

This function approximates a thresholding function $I[x \leq r^2]$ but with error inside the interval $[r^2 - \delta, r^2]$. By Lemma 2, a 2-layer ReLU network $\tilde{f}$ with size 2 can represent $f(x)$. Then $\tilde{f} \circ \tilde{L}$ is a ReLU network with depth 5 and size $(d-1)s_{11} + s_2 + s_{31} + s_{32} + 4$, such that

$$\sup_{\mathbf{x}} |f \circ L(\mathbf{x}) - \tilde{f} \circ \tilde{L}(\mathbf{x})| \leq \epsilon_1, \tag{A.20}$$

with letting $\epsilon_1 = \epsilon_1/d$.

Let $h(\mathbf{x}) = \tilde{f} \circ \tilde{L}(\mathbf{x})$. We claim that $h(\mathbf{x})$ is the desired network with depth 5 and size

$$(d-1)s_{11} + s_2 + s_{31} + s_{32} + 4$$
$$= \frac{d}{\epsilon_1}(4(d-1)(R+r)^2 + 8r^2 + \frac{r}{\sqrt{R-r}}) + 9 \tag{A.21}$$
$$= O(\frac{d^2}{\epsilon_1}).$$

To finalize our proof, we just need to bound the error $||h(\mathbf{x}) - I_{T_1}(\mathbf{x})||_{L_2(\mu)}$. The proof follows proof of Proposition 1. The error is divided into two parts and is bounded separately. The only difference is we define $S_\delta$ to be $S_\delta = \{\mathbf{x} \in \mathbb{R}^3 : r^2 - \delta \leq z^2 + (\sqrt{x^2 + y^2} - R)^2 \leq r^2\}$, such that

$$\int_{S_\delta} \mu(\mathbf{x})d\mathbf{x} \leq \epsilon_2. \tag{A.22}$$

We can get $I_1 \leq \epsilon_2$, and $I_2 \leq \epsilon_1$. Let $\epsilon_1 = \epsilon_2 = \epsilon/2$, we have

$$||h(\mathbf{x}) - I(\mathbf{x})||_{L_2(\mu)} \leq \epsilon. \tag{A.23}$$

And $h$ has size at most $\frac{2d}{\epsilon}(4(d-1)(R+r)^2 + 8r^2 + \frac{r}{\sqrt{R-r}}) + 9$. $\qquad\square$

### A.4 Approximating Topological Representatives

After getting the size arising from fundamental manifolds, we proceed to study the combination of them. We start by proving the representative property.

**Lemma 1** (Representative Property). *If $\mathcal{M} \subset \mathbb{R}^D$ is a solid manifold, then there exists a manifold $\mathcal{M}' \subset \mathbb{R}^d (d \leq 3)$ that is homeomorphic to $\mathcal{M}$, where $\mathcal{M}'$ is constructed by a finite set of fundamental manifolds $\{\mathcal{F}_i\}_{i=1}^m$ via set union and subtraction. $m \leq \beta(\mathcal{M})$ is a constant integer. We term $\mathcal{M}'$ as the **topological representative** of $\mathcal{M}$.*

*Proof.* When $d = 2$, $\mathcal{M}$ is a 2-dimensional solid manifold. By the definition of solid manifolds, the boundary $\partial\mathcal{M}$ of $\mathcal{M}$ is a closed 1-manifold. According to the classification theorem of closed 1-manifold, $\partial\mathcal{M}$ is homeomorphic to either:

1. A circle $S^1$,

2. A disjoint union of a finite number of circles.

Note that a circle is the boundary of a disk, i.e., $S^1 = \partial B^2$. Next we check the disjoint union of $m$ circles $S_1^1, S_2^1, ..., S_m^1$. Every circle $S_i^1$ is a boundary of a disk $B_i^2$. The disjoint property ensures that for any pair $i, j$, only one of the three situations can hold, $B_i^2 \cap B_j^2 = \emptyset$, $B_i^2 \subset B_j^2$ or $B_j^2 \subset B_i^2$.

**(C1):** If $B_i^2 \cap B_j^2 = \emptyset$, then $S_i^1 \cup S_j^1 = \partial(B_i^2 \cup B_j^2)$,

**(C2):** if $B_j^2 \subset B_i^2$, $S_i^1 \cup S_j^1 = \partial(B_i^2 \setminus B_j^2)$,

**(C3):** if $B_i^2 \subset B_j^2$, $S_i^1 \cup S_j^1 = \partial(B_j^2 \setminus B_i^2)$.

Every operation of (C1) introduces an additional 0-dimensional Betti number $\beta_0$, and (C2) or (C3) introduces an additional 1-dimensional Betti number $\beta_1$. In the light of this, the total number of disks $m = \beta_0(\mathcal{M}) + \beta_1(\mathcal{M}) = \beta$. Now we have proved the proposition in the case of $d = 2$.

For $d = 3$, similarly $\partial \mathcal{M}$ is a closed oriented 2-manifold. Based on the classification of closed oriented 2-manifolds, $\partial \mathcal{M}$ is homeomorphic to one or the disjoint union of some members from the following two classes:

1. A sphere $S^2$,

2. A genus-$g$ torus $\partial T_g$.

Note that we use $T_g$ to represent the solid genus-g torus, so $\partial T_g$ is the genus-g torus. A genus-g torus is the connected sum of $g$ tori. In another word, there exists $g$ solid tori, such that $T_g$ is the union of these $g$ solid tori. With $S^2 = \partial B^3$ and a similar discussion as $d = 2$, we know that $\mathcal{M}$ can be obtained by union or set subtraction of a number $m$ of balls and solid tori. Since a solid torus has $\beta_0 = 1$ and $\beta_1 = 1$, $m \leq \beta_0(\mathcal{M}) + \beta_1(\mathcal{M}) + \beta_2(\mathcal{M}) = \beta$. $\qquad \square$

**Lemma 7** (Manifold Union and Subtraction). *$\mathcal{M}_1$ and $\mathcal{M}_2$ are two manifolds in $\mathbb{R}^d$. $I_{\mathcal{M}_1}$ can be approximated by a ReLU network $h_1$ with depth $d_1 + 1$ and size at most $s_1$ with error $R(h_1) < \epsilon_1$, $I_{\mathcal{M}_2}$ can be approximated by a ReLU network $h_2$ with depth $d_2 + 1$ and size at most $s_2$ with error $R(h_2) < \epsilon_2$. Then $I_{\mathcal{M}_1 \cup \mathcal{M}_2}$ and $I_{\mathcal{M}_1 \setminus \mathcal{M}_2}$ can be approximated within error $\epsilon_1 + \epsilon_2$ by a ReLU network with depth at most $\max\{d_1, d_2\} + 2$ and size at most $s_1 + s_2 + 2$.*

*Proof.* We represent $I_{\mathcal{M}_1 \cup \mathcal{M}_2} = I_{x>0} \circ (I_{\mathcal{M}_1} + I_{\mathcal{M}_2})$ by a threshold function $I_{x>0}$. The threshold function can be approximate by a function

$$f(x) = \begin{cases} 0, & x \leq 0 \\ \frac{x}{\delta}, & x \in (0, \delta) \\ 1, & x \geq \delta. \end{cases} \tag{A.24}$$

with errors only in $(0, \delta)$. $f$ can be represented by a 2-layer ReLU network $h_f$ with size 2. Then if let $h = h_f \circ (h_1 + h_2)$, according to Lemma 4 and 3, $h$ is a neural network with depth $\max\{d_1, d_2\} + 2$ and size $s_1 + s_2 + 2$. Then we bound the error

$$
\begin{aligned}
||h - I_{\mathcal{M}_1 \cup \mathcal{M}_2}||_{L_2(\mu)} &= ||h_f \circ (h_1 + h_2) - I_{x>0} \circ (I_{\mathcal{M}_1} + I_{\mathcal{M}_2})||_{L_2(\mu)} \\
&\leq ||h_f \circ (h_1 + h_2) - f \circ (I_{\mathcal{M}_1} + I_{\mathcal{M}_2})||_{L_2(\mu)} \\
&\quad + ||f \circ (I_{\mathcal{M}_1} + I_{\mathcal{M}_2}) - I_{x>0} \circ (I_{\mathcal{M}_1} + I_{\mathcal{M}_2})||_{L_2(\mu)} \\
&\leq ||h_f \circ (h_1 + h_2) - h_f \circ (I_{\mathcal{M}_1} + I_{\mathcal{M}_2})||_{L_2(\mu)} \\
&\leq ||h_f||_\infty ||(h_1 + h_2) - (I_{\mathcal{M}_1} + I_{\mathcal{M}_2})||_{L_2(\mu)} \\
&\leq ||(h_1 + h_2) - (I_{\mathcal{M}_1} + I_{\mathcal{M}_2})||_{L_2(\mu)} \\
&\leq \epsilon_1 + \epsilon_2
\end{aligned}
\tag{A.25}
$$

Similarly, $I_{\mathcal{M}_1 \setminus \mathcal{M}_2}$ can be represented by $I_{x>0} \circ (I_{\mathcal{M}_1} - I_{\mathcal{M}_2})$. Following the same discussion, we can have the proposition proved.

$\qquad \square$

**Theorem 1** (Complexity Arising from Topology). *Suppose $\mathcal{M}'$ is the topological representatives of a solid manifold. Given $\epsilon > 0$, there exists a ReLU network $h : \mathbb{R}^d \rightarrow \mathbb{R}$ ($d \leq 3$) with depth at most $O(\log \beta)$ and size at most $O(\frac{d^2 \beta^2}{\epsilon})$, that can approximate the indicator function $I_{\mathcal{M}'}$ with error $R(h) \leq \epsilon$ for any continuous distribution $\mu$ over $\mathbb{R}^d$. $\beta$ is the topological complexity of $\mathcal{M}'$.*

*Proof.* Since $\mathcal{M}'$ is a topological representative, according to Lemma 1, there is a set of fundamental manifolds $\{\mathcal{F}_1, \mathcal{F}_2, .., \mathcal{F}_m\}$, such that $\mathcal{M}'$ can be obtained from these $m$ fundamental manifolds by set nion and subtraction. According to proposition 7, $I_{\mathcal{F}_1 \bowtie \mathcal{F}_2 \bowtie .. \bowtie \mathcal{F}_m}$ can be approximated by a ReLU network $h$ with depth at most $\max\{d_1, d_2, ..., d_m\} + \log m$ and size at most $\sum_{i=1}^{m} s_i + \log m$, with error $R(h) \leq \sum_{i=1}^{m} \epsilon_i$. $\bowtie$ is either set union or subtraction. Then according to Proposition 1 and 2, $s_i \sim O(d^2/\epsilon_i)$, $d_i \sim O(1)$ and take $\epsilon_i$ to be all the same for all $i = [m]$. Let $\epsilon = m\epsilon_i$ and note that $m \leq \beta$. We have $h$ has depth at most $O(\log \beta)$ and size at most $O(\frac{d^2\beta^2}{\epsilon})$, and can approximate $I_{\mathcal{M}'}$ with error $R(h) \leq \epsilon$. □

## A.5 OVERALL COMPLEXITY

We present a result from Gonzalez-Diaz et al. (2019), which gives a bound of network size to represent a simplicial map.

**Proposition 6** (Adapted from Theorem 4 in Gonzalez-Diaz et al. (2019)). *Let us consider a simplicial map $\phi_c : |K| \to |L|$ between the underlying space of two finite pure simplicial complexes $K$ and $L$. Then a two-hidden-layer feed-forward network $\mathcal{N}_\phi$ such that $\phi_c(x) = \mathcal{N}_\phi(x)$ for all $x \in |K|$ can be explicitly defined. The size of $N_f$ is $D + d + k(D + 1) + l(d + 1)$, where $D = dim(|K|)$ and $d = dim(|L|)$, $k$ and $l$ are the number of simplices in $K$ and $L$, respectively.*

**Theorem 2** (Main Theorem). *Let $\mathcal{M} = \mathcal{M}_1 \bigcup \mathcal{M}_0 \subset \mathbb{R}^D$ be a $d$-dimensional solid manifold. $\mathcal{M}_1$ and $\mathcal{M}_0$ are two disjoint sub-manifolds of $\mathcal{M}$ representing two classes and both are also solid. The reach of $\mathcal{M}$ is $\tau$ and the total Betti number of $\mathcal{M}_1$ is $\beta$. Given a training set $\{(\mathbf{x}_i, y_i)|\mathbf{x}_i \in \mathcal{M}, y_i \in \{0, 1\}\}_{i=1}^{n}$, where $\mathbf{x}_i$ are sampled i.i.d. from $\mathcal{M}$ by a uniform distribution, and $y_i = I_{\mathcal{M}_1}(\mathbf{x}_i)$. For any $\delta > 0$, if inequality (6) holds, then for any $\epsilon > 0$, there exists a ReLU network $g$ with depth at most $O(\log \beta + d \log \frac{1}{\tau} + \log \log \frac{1}{\tau\delta})$ and size at most $O(\frac{d^2\beta^2}{\epsilon} + \tau^{-d^2/2} \log^{d/2} \frac{1}{\tau\delta} + D\tau^{-d} \log \frac{1}{\tau\delta})$, such that*

$$P(R(g) \leq \epsilon) > 1 - \delta, \tag{A.26}$$

*where $R(g) = \int_{\mathcal{M}} (g - I_{\mathcal{M}_1})^2 \mu(\mathbf{x}) d\mathbf{x}$ with any continuous distribution $\mu$.*

*Proof.* Since $\mathcal{M} = \mathcal{M}_1 \cup \mathcal{M}_0$ is a solid manifold, it has a topological representative $\mathcal{M}' = \mathcal{M}'_1 \cup \mathcal{M}'_0 \in \mathbb{R}^d$, where $\mathcal{M}'_1$ and $\mathcal{M}'_0$ are topological representatives of $\mathcal{M}_1$ and $\mathcal{M}_2$, respectively.

The proof follows by first constructing simplicial approximations $|K|$ and $|L|$ of $\mathcal{M}$ and $\mathcal{M}'$, respectively. Then we represent a simplicial homeomorphism $\phi : |K| \to |L|$ by a neural network $N_\phi$, where $K$ is constructed from $\mathcal{M}$ and $L$ from $\mathcal{M}'$. Built on the top of this, a projection from $\mathcal{M}$ to its similicial approximation $|K|$ is represented by another network $N_p$. The overall network can be constructed by $g = h \circ N_\phi \circ N_p$. Note that $h$ is the function to approximate $I_{\mathcal{M}'_1}$, but the data after projection and homeomorphism is from $|L_1|$. There should be an error in this approximation. However, we will show that by using the true risk, having $|L_1| \subseteq \mathcal{M}'_1$ will make sure $||I_{|L_1|} - I_{\mathcal{M}'_1}||_{L_2(\mu')} = 0$. We move ahead by first constructing the network $g$, and then bound the approximation error.

**Network Construction.** Given $\mathcal{M}$ is a compact submanifold of $\mathbb{R}^D$ and $\mathbf{x}_i$ are sampled according to a uniform distribution, by Proposition 3, for all $0 < r < \tau/2$ and $n > \lambda_1(log(\lambda_2) + log(\frac{1}{\delta}))$ $(n \sim O(\tau^{-d} \log(1/\tau\delta)))$, $U = \bigcup_i B_r^D(\mathbf{x}_i)$ has the same homology as $\mathcal{M}$ with probability higher than $1 - \delta$. Note that every $B_r^D(\mathbf{x}_i)$ is contractible because $r \leq \tau$. Therefore by the nerve theorem (Edelsbrunner & Harer, 2022), the nerve of $U$ is homotopy equivalent to $\mathcal{M}$. Note that $U$ is a collection of $\epsilon$-balls. The nerve of $U$ is the Čech complex, which is an abstract complex constructed as $\check{C}ech(r) = \{\sigma \subseteq X | \bigcap_{\mathbf{x} \in \sigma} B_r(\mathbf{x}) \neq 0\}$. But since the dimension of $\mathcal{M}$ is $d$, it suffices to only consider simplices with dimension $\leq d$. Delaunay complex is such a geometric construction that limits the dimension of simplices we get from a nerve. And in the other hand, we also do not want to lose the radius constraint. Here we construct the Alpha complex, a sub-complex of the Delaunay complex. It is constructed by intersecting each ball with the corresponding Voronoi cell, $R_{\mathbf{x}}(r) = B_r(\mathbf{x}) \cap V_{\mathbf{x}}$. The alpha complex is defined by

$$\text{Alpha}(r) = \{\sigma \subseteq X | \bigcap_{\mathbf{x} \in \sigma} R_{\mathbf{x}}(r) \neq 0\}. \tag{A.27}$$

Based on the construction, Alpha$(r)$ also has the same homotopy type as $U$. Bern et al. (1995) provided the number of simplices in a Delaunay complex of $n$ vertices is bounded by $O(n^{\lceil d/2 \rceil})$.

Since the Alpha complex is a sub-complex of Delaunay complex, the number of simplices in Alpha$(r)$ is also bounded by

$$O(n^{\lceil d/2 \rceil}) = O(\tau^{-d^2/2} \log^{d/2} \frac{1}{\tau\delta}) \tag{A.28}$$

Denote $K = \text{Alpha}(r)$.

We claim that there exists a a vertex map $\phi : \mathbf{x}_i \to \mathbf{x}_i'$ for $i = 1, ..., n$, such that with probability higher than $1 - \delta$, $U' = \bigcup_i B_{r'}^d(\mathbf{x}_i')$ has the same homology of $\mathcal{M}$. We prove this claim at the after the proof. We can construct an alpha complex from $\{\mathbf{x}_i'\}_{i=1}^n$ in a similar way, $L = \text{Alpha}(r)$. The number of simiplices is also bounded by $O(\tau^{-d^2/2} \log^{d/2} \frac{1}{\tau\delta})$.

$\phi$ can be extended to a simiplicial map $\phi : |K| \to |L|$ by

$$\phi(\mathbf{x}) = \sum_{i=1}^n b_i(\mathbf{x}) \phi(\mathbf{x}_i). \tag{A.29}$$

The map $b_i : |K| \to \mathbb{R}$ maps each point to its $i$-th barycentric coordinate. According to Proposition 6, there exists a ReLU network $N_\phi$ with depth 4 and size $O(\tau^{-d^2/2} \log^{d/2} \frac{1}{\tau\delta})$, such that $\phi(\mathbf{x}) = N_\phi(\mathbf{x})$ for all $\mathbf{x} \in |K|$.

Next we construct a network $N_p$ that projects $\mathcal{M}$ to its simplicial approximation $|K|$. The point is projecting $\mathbf{x} \in \mathcal{M}$ to its closest simplex $\sigma_\mathbf{x}$. According to the proof of theorem 3 in Schonsheck et al. (2019), such projection can be represented as a neural network $N_p$ with depth at most $\log n + 1$ and size at most $O(nD)$. Lastly, by Theorem 1, a neural network $h$ with depth at most $O(\log \beta)$ and size at most $O(\frac{d^2 \beta^2}{\epsilon_1})$ can approximate $I_{\mathcal{M}_1'}$ with error $R(h) \le \epsilon_1$. And by Lemma 3, $g = h \circ N_\phi \circ N_p$ has depth at most $O(\log(n\beta))$ and size $O(\frac{d^2 \beta^2}{\epsilon_1} + \tau^{-d^2/2} \log^{d/2} \frac{1}{\tau\delta} + nD)$. Given $n \sim O(\tau^{-d} \log(1/\tau\delta))$, $g$ has depth at most $O(\log \beta + d \log \frac{1}{\tau} + \log \log \frac{1}{\tau\delta})$ and size at most $O(\frac{d^2 \beta^2}{\epsilon_1} + \tau^{-d^2/2} \log^{d/2} \frac{1}{\tau\delta} + D\tau^{-d} \log \frac{1}{\tau\delta})$. Note that the probability of the existence for such network is larger than $(1 - \delta)^2 = 1 - 2\delta + \delta^2 > 1 - 2\delta$. We let $\delta = 2\delta$, such that with probability larger than $1 - \delta$, neural network $g$ exists and $g$ has depth at most $O(\log \beta + d \log \frac{1}{\tau} + \log \log \frac{1}{\tau\delta})$ and size at most $O(\frac{d^2 \beta^2}{\epsilon_1} + \tau^{-d^2/2} \log^{d/2} \frac{1}{\tau\delta} + D\tau^{-d} \log \frac{1}{\tau\delta})$.

**Bounding Approximation Error.** Now it is time to bound the approximation error $R(g)$. We split $R(g)$ into two parts.

$$\begin{aligned} R(g) &= ||g - I_{\mathcal{M}_1}||_{L_2(\mu)} \\ &\le ||h \circ N_\phi \circ N_p - I_{\mathcal{M}_1'} \circ \phi \circ N_p||_{L_2(\mu)} + ||I_{\mathcal{M}_1'} \circ \phi \circ N_p - I_{\mathcal{M}_1}||_{L_2(\mu)} \\ &= I_1 + I_2. \end{aligned} \tag{A.30}$$

We first show that $I_2 = 0$. Note that $N_p : \mathcal{M} \to |K|$, and $\phi : |K| \to |L|$. We claim that for $\mathbf{x} \in \mathcal{M}_1$, $\phi \circ N_p(\mathbf{x}) \in L_1$ and if $\mathbf{x} \in \mathcal{M}_0$, $\phi \circ N_p(\mathbf{x}) \in L_0$. This is true because $\mathcal{M}_1$ and $\mathcal{M}_2$ are disjoint and $L$ is homotopy equivalent to $\mathcal{M}$. Consequently, $I_{L_1} \circ \phi \circ N_p = I_{\mathcal{M}_1}$.

Now it suffices to show $||I_{L_1} - I_{\mathcal{M}_1'}||_{L_2(\mu')} = 0$. Note that $\mu'$ is a distribution supported on $|L|$, it can be naturally extended to $\mathcal{M}'$ by set $\mu'(\mathbf{x}') = 0$ if $\mathbf{x}' \in \mathcal{M}'$ but $\mathbf{x}' \notin |L|$. Although this way $\mu'$ may not be continuous, Theorem 1 still holds for it, because the claim, $\exists \delta$ such that $\int_{S_\delta} \mu'(\mathbf{x}')d\mathbf{x}' \le \epsilon$, still holds.

The term $||I_{L_1} - I_{\mathcal{M}_1'}||_{L_2(\mu')}$ is not likely to be zero because there are points $\mathbf{x}' \in |L_1|$ but $\mathbf{x}' \notin \mathcal{M}_1'$, which will raise error. Note that points $\mathbf{x}' \in \mathcal{M}_1'$ but $\mathbf{x}' \notin |L_1|$ will not cause any error because $\mu'(\mathbf{x}') = 0$. However, the topological representative $\mathcal{M}'$ is flexible in a way that we can adjust its radius. We claim that one can extent the boundary of $\mathcal{M}_1'$, such that $|L_1| \subset \mathcal{M}_1'$ and $\mathcal{M}_1'$ is still the topological representative of $\mathcal{M}_1$. We prove this in Claim 2. After the expansion, $|L_1| \subseteq \mathcal{M}_1'$ because $|L_1|$ is constructed within a $r$-cover of the old $\mathcal{M}_1'$. As a conclusion, $||I_{L_1} - I_{\mathcal{M}_1'}||_{L_2(\mu')} = 0$.

Now we settle $I_1$ with the new $\mathcal{M}_1'$. Given $N_\phi$ is an exact representation of the simplicial map $\phi$,

$$I_1 = ||h - I_{\mathcal{M}_1'}||_{L_2(\mu')}. \tag{A.31}$$

As we discussed, Theorem 1 still holds for $\mu'$. Therefore $I_1 \le \epsilon_1$. Combined together, we have

$$R(g) \le \epsilon_1. \tag{A.32}$$

Note that this inequality holds only with probability larger than $1 - \delta$ because that is the probability we successfully recover the homology of $\mathcal{M}$ by the training set and construct a simplicial homeomorphism. $\qquad\square$

**Claim 1.** *$\mathcal{M} \in \mathbb{R}^D$ is a d-dimensional solid manifold. Suppose there exists a set $\{\mathbf{x}_i \in \mathcal{M}\}_{i=1}^n$ and radius r, such that $U = \bigcup_i B_r^D(\mathbf{x}_i)$ is a cover of $\mathcal{M}$ and has the same homology. Then there exists a $\mathcal{M}' \in \mathbb{R}^d$ that is a topological representative of $\mathcal{M}$. Denote the homeomorphism between them $f$. Then with probability larger than $1 - \delta$, $U' = \bigcup_i B_{r'}^d(f(\mathbf{x}_i))$ has the same homology as $\mathcal{M}$.*

*Proof.* We let

$$c(r, \tau, \mathcal{M}) = \frac{vol(\mathcal{M})}{(cos^d\theta_1)vol(B_{r/4}^d)} \left( \log \frac{vol(\mathcal{M})}{(cos^d\theta_2)vol(B_{r/8}^d)} + \log \frac{1}{\delta} \right), \qquad (A.33)$$

where $\theta_1 = \arcsin \frac{r}{8\tau}$, $\theta_2 = \arcsin \frac{r}{16\tau}$ and $0 < r < \tau/2$. Given a set $\{f(\mathbf{x}_i)\}_{i=1}^n$, apply proposition 3 to $\mathcal{M}'$. If

$$n > c(r', \tau', \mathcal{M}'), \qquad (A.34)$$

then with probability $1 - \delta$, $U' = \bigcup_i B_{r'}^d(f(\mathbf{x}_i))$ has the same homology as $\mathcal{M}'$, with $r' < \tau'/2$.

Note that $n$ already satisfy that $n > c(r, \tau, \mathcal{M})$, it suffices to show $c(r, \tau, \mathcal{M}) > c(r', \tau', \mathcal{M}')$. Since $\mathcal{M}'$ is one of topological representatives of $\mathcal{M}$, we can always choose the radius of the fundamental members in $\mathcal{M}'$ and choose the distance between $\mathcal{M}'_1$ and $\mathcal{M}'_2$, to make sure that $\tau' > \tau$ and $vol(\mathcal{M}') < vol(\mathcal{M})$. Hence, we can choose $r$ and $r'$, such that $B_{r'}^d > B_r^d$. With the same $\delta$, we have proved that $c(r, \tau, \mathcal{M}) > c(r', \tau', \mathcal{M}')$.

$\qquad\square$

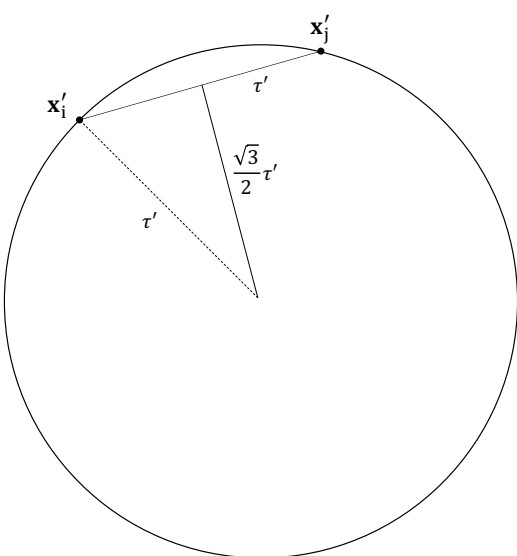

Figure A.1: The worst case in 2 dimension.

**Claim 2.** *$\mathcal{M}'_1$ is a topological representative of $\mathcal{M}_1$, $|L_1|$ is the alpha complex constructed from the $r'$-cover $U' = \bigcup_i B_{r'}^d(\mathbf{x}'_i)$, where $r' < \tau'/2$. There is a way to extend $\mathcal{M}'_1$, such that $|L_1| \subset \mathcal{M}'_1$, and $\mathcal{M}'_1$ is still a topological representative of $\mathcal{M}_1$.*

To prove this claim, we need to think about this question: what is the worst case among $|L_1| \not\subset \mathcal{M}'_1$? Or in another word, what is their largest distance could be?

Note that $|L_1|$ is constructed from a $r'$-cover, in a way that there will be an edge only if two balls have intersection. Hence, for any edge $(\mathbf{x}'_i, \mathbf{x}'_j) \in |L_1|$, the length $l_{ij}$ of it satisfies $l_{ij} < 2r' < \tau'$. And notice $\tau'$ is also the smallest radius that an inner hold can hold. So the worst case goes with, an inner ball/circle with two center points $\mathbf{x}'_i$ and $\mathbf{x}'_j$ on it, where $l_{ij} = \tau'$. The boundary of $\mathcal{M}'_1$ can be

categorised into exterior boundary $\partial^+ \mathcal{M}_1'$ and interior boundary $\partial^- \mathcal{M}_1'$. The difference between $\mathcal{M}_1'$ and $|L_1|$ can only occurs in the interior boundary $\partial^- \mathcal{M}_1'$. We prove that, under the worst cases, one can extend the interior boundary $\partial^- \mathcal{M}_1'$ inward, such that $|L_1| \subset \mathcal{M}_1'$.

In 2-dimension, illustrated in Figure A.1. We can easily calculate that the furthest distance between $(\mathbf{x}_i', \mathbf{x}_j')$ and the circle is $\Delta = (1 - \frac{\sqrt{3}}{2})\tau'$. So if we shrink the circle by $\Delta$, $(\mathbf{x}_i', \mathbf{x}_j')$ will be inside the manifold. That is, we let $\mathcal{M}_1' = \mathcal{M}_1' \cup \left( \bigcup_{\mathbf{x}' \in \partial^- \mathcal{M}_1'} B_\Delta^d(\mathbf{x}') \right)$. Note $\mathcal{M}_1'$ is still the topological representative of $\mathcal{M}_1$ because topology does not change and it still can be composed by disks.

In 3-dimension, the only difference is, we need to calculate the largest distance between an equilateral triangle $(\mathbf{x}_i', \mathbf{x}_j', \mathbf{x}_k')$ with side length $\tau'$ and its circumscribed sphere with radius $\tau'$. This distance is also easy to be calculated as $\frac{\tau'}{2}$. If let $\mathcal{M}_1' = \mathcal{M}_1' \cup \left( \bigcup_{\mathbf{x}' \in \partial^- \mathcal{M}_1'} B_{\tau'/2}^d(\mathbf{x}') \right)$, then $|L_1| \subset \mathcal{M}_1'$.

