# OpenReview forum: "Neural Network Expressive Power Analysis Via Manifold Topology"
_ICLR.cc/2024/Conference — Submitted to ICLR 2024_

### Official Review · Reviewer_mSxa · 2023-10-13

**Soundness:** 4 excellent
**Presentation:** 3 good
**Contribution:** 2 fair
**Rating:** 5
**Confidence:** 4

**Summary:**

The paper presents theory in regards to the expressivity of a multi-layer perceptron (MLP) in classifying between two classes, when each class is drawn from a smooth low-dimensional submanifold of a Euclidean space $\mathbb{R}^D$. In a restricted setting where topological theory is more rich (intrinsic dimension at most 3, compact, orientable, and have boundary), the authors make an explicit construction of a neural network classifier, where the network size is bounded by terms of separated topological and geometric complexity. The authors then finally present experimental results to demonstrate the trend between topological complexity (measured through Betti numbers) and needed network size to sufficiently classify the data.

**Strengths:**

- Visual aids for mathematical ideas were well designed and placed, helping greatly with readability.
- The theory is very well done and written in a way that highlights the connection between Betti numbers and neural network classification.

**Weaknesses:**

(roughly ordered from least to most important)
- A small handful of relatively minor typos (e.g. "quaratically" under main theorem, Definition 2 $\mathcal{M} \subset \mathbb{R}^D$ instead of $\in \mathbb{R}^D$, "fundamental" quote formatting on page 5). While needed to be fixed, these make 0 impact on my overall review.
- The term "classification" is overloaded in this paper, and the expected ICLR reader would likely be mislead by some of the writing. Namely, there are times when the author is talking about geometric manifold classification problems, where the goal is determining if one (or a set of) manifold(s) is homeomorphic to another. This is fundamentally different from neural networks classifying two classes. Since explicit clarification and distinction is missing, there are a number of misleading parts of the text. For example, the statement "classifying manifolds beyond three dimensions is known to be equivalent to the word problem on groups [...] and is undecidable" would lead the standard reader to think that this paper's restriction to at most 3-dimensional manifolds actually retains generality for neural network classification theory, but indeed this statement is talking about geometric manifold classification, not neural network classification.
- The authors' claim that their "result reveals for the first time how the topology, as a global structural characterization of data manifold, affects the network expressiveness" is misleading; while this analysis related to specifically Betti numbers and neural network expressiveness, this work is not the first to relate "structural characterization of data manifold" to nextwork expressiveness. While their related works section does cover a good breadth of related work in the intersection of topology and deep learning, one work in particular was missed that is quite close in setting and goal to this paper, namely an ICLR 2021 paper titled ["Deep Networks and the Multiple Manifold Problem"](https://openreview.net/pdf?id=O-6Pm_d_Q-). This referenced paper is of course not a strict superset (e.g. "Deep Networks[...]" only focuses on 1-dimensional manifolds), but Buchanan et al. do lay down a lot of fundamental theory on relating different sizes of a network (e.g. width and depth) to the complexity of the data manifolds and the distributions over them that data is drawn from, much of which this paper misses and could benefit from integrating in terms of practicality.
- I am having a harder time seeing how this theory applies to deep learning, outside of the analysis being over MLP's. If the goal is a new method for constructing MLP's from data, the paper's main neural network construction seems to rely on a decomposition of a data manifolds into a union of fundamental manifolds (e.g. ball, torus), which is hardly known or even computable from fixed, finite samples of practical data manifolds, and likewise does not characterize the type of functions typical MLP's fit. If the goal is to highlight the expected relationship between the sum of Betti numbers and needed network size to classify data, then the experiments shown are not nearly thorough enough to demonstrate this, not only due to unrealistic settings (incredibly high sampled data density, low dimensional, etc.), but because the only trend plotted was over a single family of manifolds (g-genus tori), which alongside its Betti numbers sum has many other geometric properties growing that could explain the expressivity trend. It is thus not clear from these demonstrations why it is meaningful to focus on the Betti numbers.

**Questions:**

- Suppose someone has two classes of rotated-MNIST, the 5's and the 9's (for our rotated-MNIST each image from MNIST is rotated 2*pi radians exclusive at, say, fidelity 10; so a single MNIST training class goes from 6,000 to 60,000 data points). They want to train an MLP-classifier between these two classes. How would they use the theory and results introduced in this paper to inform their initial choice of the width and depth of the MLP they start off with?
- In the study of data manifolds and MLP classifiability, why should the research community focus analysis on Betti numbers for data manifolds instead of other previously studied measures of extrinsic geometry?

---

> ### Author Response · Authors · 2023-11-22
> **Author Response to Reviewer mSxa**
>
> Thank you for your comprehensive and insightful review of our paper. We appreciate your detailed feedback and have addressed the points raised to enhance the quality and clarity of our work.
>
> **Q1. How can this theory be applied to deep learning?**
>
> Thank you for your detailed feedback and for highlighting key concerns about the practical applicability of our theoretical findings. We value your insights and would like to address these concerns in the context of our study’s objectives and contributions.
>
> Our primary goal is not to devise a method for constructing MLPs from data. Rather, our research aims to deepen the theoretical understanding of how the topology and geometry of data manifolds influence neural network size. We accomplish this by establishing an upper bound for network size required for classifying data from different manifolds. This approach is consistent with numerous theoretical studies in the field that focus on the expressive power of neural networks. Similar works like [1, 2] have provided various upper bounds for neural network sizes in approximating distinct function families, and while these results might not directly apply to datasets, they are invaluable for enhancing our fundamental comprehension of neural networks.
>
> The insights from our theoretical exploration shed light on the intricate relationship between the complexity of data manifolds and the requisite size of neural networks for effective classification. This enhanced understanding has significant potential to guide future research in deep learning, especially regarding the optimization of network architectures based on the characteristics of the underlying data.
>
> Regarding the necessity and relevance of topological complexity and Betti numbers in our study, we invite you to refer to Q1 and Q3 in our summary.
>
> In essence, while our results are rooted in theory, they form an integral part of the evolving understanding of neural networks, laying a foundation for further empirical exploration and practical application in the field of deep learning.
>
> **Q2. About the claim of contributions**
>
> We sincerely thank the reviewer for directing our attention to the paper [3], which was not previously included in our references. While this paper is indeed relevant, it does not diminish the novelty of our claim. Our work uniquely demonstrates for the first time how the topology, as a global structural characterization of data manifolds, impacts neural network expressiveness.
>
> The study in [3] focuses on a specific category of manifolds: smooth, regular simple curves. These manifolds are characterized by being connected and non-self-intersecting, implying a uniform topology with only one connected component. Consequently, their analysis does not necessitate the inclusion of topological descriptors.
>
> In contrast, our research encompasses a broader spectrum of manifolds with diverse topological features. The incorporation of topological complexity into our upper bound represents a pioneering effort in illustrating how varying topological characteristics of data manifolds influence the size and structure of neural networks. This aspect of our work stands as a distinct and significant contribution to the field, highlighting manifold structures and their relationship with network expressiveness.
>
> **Q3. Ambiguity of the term ‘classification’.**
>
> We appreciate your attention to the potential ambiguity surrounding the term "classification" as it pertains to both the manifold classification theorem and classification problems in machine learning. In response to your insightful comment, we have carefully revised our manuscript to include clear distinctions and explanations whenever we refer to the manifold classification theorem. This ensures that our usage of the term is precise and unambiguous within the context of our discussion.
>
> **Q4. Typos.**
>
> We are grateful to the reviewer for highlighting the typos in our manuscript. We have fixed them in the updated version.
>
> [1] Minshuo Chen, Haoming Jiang, Wenjing Liao, and Tuo Zhao. Efficient approximation of deep ReLU networks for functions on low dimensional manifolds. NeurIPS, 2019
>
> [2] Ronen Eldan and Ohad Shamir. The power of depth for feedforward neural networks. CoLT, 2016.
>
> [3] Sam Buchanan, Dar Gilboa and John Wright. Deep Networks and the Multiple Manifold Problem. ICLR 2021.

---

> > ### Comment · Reviewer_mSxa · 2023-11-22
> >
> > Thank you for your detailed response and updates to the manuscript. I have read through it again and agree that my first three comments are resolved. However, I still feel my point in my 4th bullet point is fundamental and unaddressed -- not the purpose of the paper, I agree the purpose through the updated manuscript is clear now, but how this theory is justified as a truly predictive theory of neural network size.
> >
> > I have increased my score to a 5. I would also like to note that a common complaint from other reviewers has been the restriction to *intrinsic* dimension 3, but I don't find this to be a fundamental weakness (I've also noticed I'm the only confidence >3 reviewer). Geometric theory in general dimensions builds up a lot from rich theory developed in lower dimensions, and since a lot of new geometry needs to be made for the modern data era, papers such as this that focus on lower (but still interesting) intrinsic dimension still add a great deal in the pursuit of general-dimensional geometric theories for data and deep learning.
> >
> > However, as also stated in my initial review, a fundamental limitation still remains: there is still not nearly enough empirical evidence to justify that the proposed topological properties are a strong determining factor of the needed network size. I understand this is an empirical request for a theoretical paper, but such is important when claiming a theory is predictive or giving true theoretical understanding for deep learning -- analyzing 1 family of manifolds is not nearly enough to disentangle Betti numbers from many other geometric properties also growing with this family, and thus there is no case made to justify the use of Betti numbers over other well-studied geometric properties: for example, justifying why one should research intrinsic properties like Betti numbers over extrinsic properties like extrinsic curvature, as extrinsic geometry and the embedding space will ultimately play some role in neural networks acting directly in said embedding space.
> >
> > I understand this is the last day for rebuttal -- I will try to stay active on this thread in case the authors have more experimental evidence to present, which if convincing enough I will of course increase my score further. As it stands though, while the theoretical work is interesting in its own right, I do not feel there is enough empirical evidence to justify this as a truly predictive theory for neural network size.

---

> > > ### Author Response · Authors · 2023-11-22
> > > **Author Response to Reviewer mSxa**
> > >
> > > Thank you for your thoughtful feedback and for recognizing the significance of starting our exploration from a low intrinsic dimension. We deeply appreciate your insights and the increased score you have provided.
> > >
> > > In response to your concerns about the justification of our theory regarding the influence of Betti numbers on neural network size, we acknowledge the need for more empirical evidence, as you have rightly pointed out. We'd like to address this in two key areas:
> > >
> > > 1. **Interplay of Intrinsic and Extrinsic Properties:** As highlighted in our summary response Q1, both intrinsic and extrinsic properties play crucial roles in the analysis of classification neural networks. This is because, during classification, these properties are compressed together. In our summary response Q1, we demonstrate that curvature alone is insufficient to determine the necessary size of a neural network. We provided an example where two datasets, despite having identical curvature measures, require neural networks of different sizes for effective classification. This underscores the need to consider a more comprehensive range of properties.
> > >
> > > 2. **Empirical Evidence from Related Studies:** We base our theory on existing empirical studies that demonstrate a correlation between complex Betti numbers and the size of networks required for classification. For instance, the study referenced as [1] conducts an extensive examination of synthetic data. Their findings, particularly in Figure 5, reveal that datasets with larger Betti numbers necessitate larger neural networks for comparable levels of convergence compared to those with smaller Betti numbers. Another relevant study, [2], investigates changes in Betti numbers across latent representations in neural network layers. Their experiments, which include both synthetic and real-world datasets, although focusing on a different aspect, also highlight the relationship between Betti numbers and the size of neural networks in a classification context.
> > >
> > > We believe these points reinforce the basis for our theoretical approach, and we are committed to further empirical investigations to strengthen our theory. Your feedback is invaluable to our ongoing research, and we thank you again for your constructive critique.
> > >
> > > [1] William H Guss and Ruslan Salakhutdinov. On characterizing the capacity of neural networks using algebraic topology. arXiv, 2018.
> > >
> > > [2] Gregory Naitzat, Andrey Zhitnikov, and Lek-Heng Lim. Topology of deep neural networks. JMLR, 2020.

---

### Official Review · Reviewer_ai47 · 2023-10-30

**Soundness:** 2 fair
**Presentation:** 3 good
**Contribution:** 3 good
**Rating:** 5
**Confidence:** 2

**Summary:**

The paper studies expressive power of ReLU networks. The main result makes a connection between topology of a data manifold, as evaluated by Betti numbers, and a size (depth and width) of a ReLU network sufficient for its good approximation. The paper is mostly theoretical. Empirical results demonstrate tightness of the proposed bounds for one setting.

**Strengths:**

1) Only a few papers studies connection between topological complexity of data manifold and size of deep NN.
New theoretical results establish such a connection.
2) The paper is well written and easy to follow.
3) Experiments with tori in 3D demonstrate the tightness of bounds.

**Weaknesses:**

1) The paper contains only one experiment on a synthetic dataset, which is a union of solid tori.
A typical paper from ICLR is expected to have a larger experimental section. Why not to try more complex datasets: entangled tori, nested spheres, etc. (see Naitzat, 2020).

2) In Section 3, you write "To explore the influence of manifold topology on network size, our analysis framework primarily
leverages manifold classification theorems". There seems to be an ambiguity of the term "classification". Here, you, probably mention **classification** of manifolds by their structure. While in ML, by classification we mean a separation of two manifolds in the shared ambient space, via a hyperplane of a non-linear separating surface.

3) In experimental section, the testing loss, which is of main interest in ML/AI is not studied.

For now, I think that limited experimental section is the main drawback of this manuscript (I haven't checked math carefully).

**Questions:**

1) In assumption 1, you state that M can be embedded into R^d, where d ≤ 3 and d ≤ D.
This restriction seems to be very tight, because dimensionalities of manifolds from real datasets are much higher.
Can it be relaxed?

2) Naitzat, et al. 2020 showed that the topological complexity of data manifold (and its representations in NN) decreases with depth, while in your work you find that width is more important (beta^2 vs. log(beta)). Can you explain it?

**Post rebuttal**. Thank you for the response. Only a minor part of issues were resolved. I realize that the manuscript is a theoretical one, but I would like to see more experiments, see weaknesses 1,3. For now, I prefer to remain my score unchanged.

---

> ### Author Response · Authors · 2023-11-22
> **Author Response to Reviewer ai47**
>
> Thank you for your detailed review and valuable feedback on our manuscript. We appreciate your insights and have addressed the points you raised as follows:
>
> **Q1. Limited experimental section.**
>
> We are grateful for your insightful suggestions regarding the experimental aspects of our work. Your feedback highlights important areas for potential expansion and improvement.
>
> We would like to clarify that the core contribution of our paper is the establishment of an upper bound, as detailed in our main theorem. This result stands on its own merit, grounded in rigorous mathematical proof. Therefore, the necessity of experimental validation for this specific aspect of our work is not as critical, since the theorem is proven through theoretical methods.
>
> That said, the inclusion of the experimental section was motivated by an intriguing observation: in certain settings, the impact of topological complexity, as predicted by our theorem, becomes quite apparent. We acknowledge that these findings may not be universally applicable across all settings. However, the mere existence of settings where the topological complexity bound is observed to be tight serves as a compelling illustration of our theoretical results.
>
> We understand your perspective on the breadth of experimental validation and appreciate the opportunity to consider how our work might be further enriched. We believe this discussion enhances the understanding of our work's scope and its application in various contexts.
>
> **Q2. Ambiguity of the Term "Classification".**
>
> We appreciate your attention to the potential ambiguity surrounding the term "classification" as it pertains to both the manifold classification theorem and classification problems in machine learning. In response to your insightful comment, we have carefully revised our manuscript to include clear distinctions and explanations whenever we refer to the manifold classification theorem. This ensures that our usage of the term is precise and unambiguous within the context of our discussion.
>
> **Q3. Limitation of manifold’s dimension.**
>
> Please refer to Q2 in our summary response.
>
> **Q4. Topological complexity and network depth.**
>
> Thank you for highlighting the comparison with the findings in Naitzat, et al. 2020. We would like to clarify that our results do not contradict their experimental outcomes. Firstly, our analysis indicates that the size of the neural network, in terms of the total number of neurons, is proportional to \(\beta^2\), focusing on the overall network size rather than just its width. It's a common understanding that the size of a neural network can exceed its depth, which aligns with our findings.
>
> Secondly, while Naitzat et al. 2020 [1] observed that the topological complexity of internal representations decreases with network depth, this aligns with the expectation in classification tasks. Typically, the final outputs are binary (0/1), leading to a collapse of topological complexity at the middle layers of the network. In contrast, our work specifically examines how the initial topological complexity of input manifolds influences the overall size and depth of a neural network. Our focus is on the input side of the network, rather than the evolution of topological features within the network layers.
>
> We believe these clarifications underscore the distinct yet complementary nature of our work with respect to the study by Naitzat et al. 2020.
>
> [1] Gregory Naitzat, Andrey Zhitnikov, and Lek-Heng Lim. Topology of deep neural networks.
> JMLR, 2020.

---

### Official Review · Reviewer_9Qss · 2023-11-01

**Soundness:** 3 good
**Presentation:** 3 good
**Contribution:** 3 good
**Rating:** 8
**Confidence:** 3

**Summary:**

This work looks at neural network expressive power in terms of the topological and geometric properties of the underlying data manifolds. The work establishes upper bounds on the size of the network needed to distinguish between data drawn from two manifolds as a function of their geometric and topological properties. To quantify topology, the work uses the sum of the Betti numbers (also known as the *topological complexity*) and to quantify geometry, the work uses the reach of the manifolds. Informally, the latter can be thought of as being related to curvature. The main result of the paper gives an upper bound on network depth and network size (the sum of the hidden dimensions) in terms of these two quantities. The work puts a number of assumptions on the manifolds in question, including having dimension $3$ or less and being able to be embedded in $\mathbb{R}^3$, being compact, and being orientable.

**Strengths:**

**Quality of mathematics:** While the reviewer did not have time to check all the proofs, the mathematical constructions seemed reasonable. It makes sense to start with topological complexity and reach when looking for statistics that capture topology and geometry respectively.

**Presentation:** Besides some typos and grammatical mistakes, the work was mostly well-presented. The format used for proceedings papers at machine learning conferences is not especially conducive to deep mathematical works that require some building up towards a proof, but the reviewer felt that this paper did a good job conveying the main ideas in the body even if all the proof work had to go into the Appendix. The reviewer appreciated that all the main ideas and assumptions were both motivated and clearly defined. An informal explanation of the significance of individual results was also useful.

**Experiments:** The experimental section was appreciated, and helped ground some of the results. The relationship between the experiments and the main theorem was obvious and required only minimal explanation.

**Weaknesses:**

- **Assumptions on manifolds:** The types of manifolds that are considered are relatively limited. Some of the assumptions (such as compactness) are reasonable when thinking about data manifolds. Other assumptions, such as $d$-dimensionality, where $d \leq 3$, seems somewhat limiting (for instance, [1] provided evidence that ImageNet has an intrinsic dimension between 26 and 43). That being said, other works in this research area also put constraints on some part of the set-up (or have correspondingly weaker conclusions) so this may be the price of progress.

- **Hints at the proof technique:** For papers such as this where the main contribution is a theorem, it is useful to give a high-level overview of the proof, even if the proof itself is relegated to the Appendix. This was done to some extent in this paper, but some additional details could probably be included for clarity. For instance, it seemed that some classification theorem was being used (and this presumably was what determined the choice to constrain to *solid manifolds*), but it was not clear what this was from the body of the work. Further spelling out some of these dependencies might illuminate why various assumptions were made.

### Nitpicks
- The abstract and introduction consider the “size” of the network. This should be at least informally defined (later it is revealed to be the sum hidden dimensions).
- “indication function” $\mapsto$ “indicator function”.
- There are a number of works that look at topological properties of ReLU networks that were not cited in the paper that may be worth considering (e.g., [2]).

[1] Pope, Phillip, et al. "The intrinsic dimension of images and its impact on learning." arXiv preprint arXiv:2104.08894 (2021).
[2] Grigsby, J. Elisenda, and Kathryn Lindsey. "On transversality of bent hyperplane arrangements and the topological expressiveness of ReLU neural networks." SIAM Journal on Applied Algebra and Geometry 6.2 (2022): 216-242.

**Questions:**

- The reviewer was not previously aware of the term “solid manifold”. Is this an idea developed in this work or elsewhere?
- The work references the use of manifold classification theorems, what manifold classification theorem was actually used?

---

> ### Author Response · Authors · 2023-11-22
> **Author Response to Reviewer 9Qss**
>
> We greatly appreciate your thorough review and constructive feedback on our work. Your insights have been invaluable in enhancing the clarity of our work. Below, we address the points you raised.
>
> **Q1. Assumptions on manifolds’ dimensions.**
>
> The constraint to manifolds with dimension \(d \leq 3\) may appear overly restrictive at first glance. However, it's important to emphasize that the analysis of 3-dimensional manifolds is far from trivial. These manifolds present unique complexities and require intricate theoretical approaches. For a more detailed discussion on this, please refer to Q1 in our summary response.
>
> **Q2. Suggestions on the manuscript.**
>
> We are grateful for your insightful suggestions regarding the proof sketch, typos, and references. In response, we have thoroughly revised and updated our manuscript to reflect these valuable inputs.
>
> **Q3. Is solid manifold a concept developed in this paper?**
>
> Yes, in our study, we use the term 'solid manifolds' to describe manifolds that meet our specific assumptions. This terminology is adopted for clarity and to facilitate the proof process.
>
> **Q4. What manifold classification theorem was actually used?**
>
> In the proof of Lemma 1, we use the classification theorem for closed 1-manifolds and the classification theorem for closed surfaces.

---

> > ### Comment · Reviewer_9Qss · 2023-11-22
> > **Reviewer response**
> >
> > Thanks for the clarification, interesting paper.

---

### Official Review · Reviewer_DRpb · 2023-11-01

**Soundness:** 2 fair
**Presentation:** 3 good
**Contribution:** 2 fair
**Rating:** 3
**Confidence:** 2

**Summary:**

This paper shows the relationship between the topological complexity of the data manifold and the depth and size required for expressive power of the classification network. This result indicates that the topological complexity perspective of data manifolds may be useful for network construction.

**Strengths:**

Upper bounds on the depth and size of the ReLu network are theoretically given, taking into account the phase complexity of the data manifold.

**Weaknesses:**

The settings discussed in this paper seem unrealistic from a realistic machine learning perspective.
-  It is generally recognized that data manifolds are embedded in low-dimensional manifold, but anything below 3 dimensions seems too low-dimensional. In three dimensions or less, the target is likely to be relatively simple and easy to analyze by machine learning. Therefore, even if correctly shown in these settings, many theories do not hold true in higher dimensions. In other words, this result is not recognized as a result that can be passed on a general level. On the other hand, if it does not hold in these settings, it will not hold in general. So I recognize that the discussion in this paper is positioned as a pilot study that suggests that it may be important to consider topological complexity.
- The discussion in this paper uses Betti numbers for data manifolds. However, since the data is typically given as a point cloud, the definition of the Betti number is not unique. This is the same as the discussion of filtration in persistent homology. In other words, it is not appropriate as a theory to evaluate a classification network if the results differ depending on the method used to calculate the Betti number, even for the same data.
- The assumption is that the phase of the data manifold is known, but it seems difficult to know that in advance. Are we not specifically discussing building a network, but merely discussing relationships? In other words, is the goal to suggest the value of considering topology?
- The network discussed in this paper introduces $N_p$ and $N_{\phi}$ first. However, a general classification network does not include $N_p,N_{\phi}$, nor is there a need to introduce it. Also, from the proof, the result of the main theorem seems to be strongly influenced by $N_p,N_{\phi}$. In other words, the authors have failed to show that topology considerations are important for general classification networks, the only thing that can be shown is the obvious fact that networks with the additional topology analysis functionality introduced by the author himself are affected by topology complexity.
- The assumption in this paper seems to be that the classes of classification are disjointed. Is that a realistic setting?
- This discussion seems to be overly argumentative for the purpose of simply binary classification. Perhaps it would be more appropriate to discuss this in a setting such as expressive learning.

**Questions:**

I would like you to answer the questions I wrote in the weaknesses section above. I would like to hear thoughts on how this fits into the scope of this meeting, especially since the setting is so far from a realistic machine learning setting.

---

> ### Author Response · Authors · 2023-11-22
> **Author Response to Reviewer DRpb**
>
> We thank the reviewer for the thoughtful comments. We would like to address your questions with the following discussions.
>
> **Q1. Manifolds in three dimensions or less are too easy.**
>
> The complexity of low-dimensional manifolds (3 or lower) should not be underestimated. It is not true that if a theory does not hold in low-dimensional cases, then it won’t hold in general. They differ from high-dimensional manifolds (5 or higher) in the theoretical framework. Low-dimensional manifolds often present unique complexities and challenges in analysis. Please refer to Q1 in our summary response.
>
> **Q2. Betti numbers are not accessible from point clouds.**
>
> Contrary to the belief that Betti numbers are inaccessible from point clouds, they can indeed be accurately computed from sufficiently dense samples. For more information, please see our detailed response to Q3 in the summary.
>
> **Q3. Is the goal of this paper to suggest the value of considering topology?**
>
> Our paper emphasizes the intrinsic role of topology in classification problems. We discuss this further in response to Q1 in the summary, addressing the reviewer's concerns about the prior knowledge of the data manifold's phase.
>
> **Q4. Real neural networks do not work as the construction in the paper.**
>
> We aim to establish an upper bound on network complexity in classification scenarios. And the methodology is to prove by construction. While classification neural networks may not function as a straightforward homeomorphism followed by a classifier, the derived upper bound remains valid, suggesting that real networks could have fewer parameters. Building an upper bound on the size of neural networks to analyze its expressive power is a common approach in many theoretical papers, like [1, 2].
>
> **Q5. The assumption in this paper seems to be that the classes of classification are disjointed. Is that a realistic setting?**
>
> The assumption that classes in classification are disjoint is both intuitive and realistic, as overlapping manifolds would imply ambiguously labeled points, which is uncommon. This theoretical setting is also employed in other studies, such as [3].
>
> **Q6. It would be more appropriate to discuss this in a setting such as expressive learning.**
>
> We appreciate the suggestion to frame our discussion in the context of expressive learning. Typically, neural network expressiveness is analyzed through function approximation under assumptions of continuity or smoothness, as in [2]. However, classification involves discontinuous real functions, presenting greater analytical challenges due to their less favorable properties.
>
> [1] Ronen Eldan and Ohad Shamir. The power of depth for feedforward neural networks. CoLT, 2016.
>
> [2] Minshuo Chen, Haoming Jiang, Wenjing Liao, and Tuo Zhao. Efficient approximation of deep ReLU networks for functions on low dimensional manifolds. NeurIPS, 2019
>
> [3] Sam Buchanan, Dar Gilboa and John Wright. Deep Networks and the Multiple Manifold Problem. ICLR 2021.

---

> > ### Comment · Reviewer_DRpb · 2023-11-23
> >
> > First I would like to appreciate the author's detailed response.
> >
> > Although many of the responses were different from the intent of the question, we understood that this paper was not assuming a realistic setting, but was arguing that it was important to analyze what would happen if the training were conducted in an ideal situation. On the other hand, one of the main purposes of neural networks is usually to transform intricate data into simple manifolds. The theorem presented in this paper does not take into account anything about transformations up to solid manifolds. From this perspective, it is difficult to say that the depth of the neural network is being evaluated.
> >
> > On the other hand, we think it is meaningful to show that topology information affects the depth of the neural network. However, once converted to a set of solid manifolds, the effect on depth itself should be obvious, since most of the data complexity corresponds to the Betti number. It is also not clear whether the nonlinear transformations in the regular use case are homeomorphic, so it is not clear whether this evaluation is meaningful. In other words, what is presented in this paper makes sense only under very limited conditions and is a gap from the purpose of neural networks.
> >
> > In light of the above, I would keep score.

---

### Author Response · Authors · 2023-11-22
**Summary of Author Response to All the Reviewers - P1**

We appreciate all reviewers for their time and insightful comments. Your feedback is invaluable in enhancing the clarity and impact of our work. We revised the manuscript based on the constructive feedback and suggestions from the reviewers. We have uploaded the revised version to reflect the modifications (highlighted in blue).

We understand the emphasis placed on the practical applications of our theoretical framework. However, we would like to highlight the foundational importance of the theory itself. Theoretical advancements often precede and guide practical applications, laying the groundwork for future empirical research and technological developments. Our work aims to contribute to this vital groundwork. Based on some common concerns, we would like to clarify the necessity and challenges of our research on manifold topology.

**Q1. Is topological complexity analysis necessary?**

Topological complexity analysis is **essential and indispensable** in **classification** problems, complementing rather than replacing geometrical analysis. Consider a dataset derived from a manifold encompassing two classes. When applying a neural network for classification, it essentially projects this manifold onto the real line $\mathbb{R}$. Ideally, we aim for the network to map the manifold distinctly onto 0 and 1. Such a process induces significant topological and geometrical transformations: topological features like connected components and loops are condensed, while geometric characteristics, including curvature, are lost. This illustrates why both topology and geometry are integral to tackling classification problems effectively.

For instance, projecting a line segment $L_1 = [0, 1]$ to a single point requires two parameters, marking the segment's ends. However, projecting a disjoint union of segments $L_2 = [0,1] \cup [3, 4]$ to a point demands four parameters, despite $L_1$ and $L_2$ having identical curvature.

Current theoretical frameworks often overlook topological complexity in determining neural network size due to two primary reasons. Firstly, many studies focus on scenarios devoid of topological alterations, where the neural network's target function is a homeomorphism. Common examples include tasks like reconstruction or dimensionality reduction, where the manifold's intrinsic properties, such as topology, remain unchanged, and only extrinsic properties influenced by the embedding space are altered. Secondly, existing research frequently concentrates on very specific objects with uniform topology. For instance, [1] is limited to balls in $\mathbb{R}^d$, and [2] focuses exclusively on smooth regular simple curves, both representing objects with a singular connected component and devoid of complex topological features.

Our work expands the scope beyond such specific cases. We explore more general manifolds, not limited to simple shapes like balls and curves. We provide an upper bound for the size and depth of neural networks necessary for effectively classifying between two classes on a manifold, accounting for the inherent topological complexities.

---

> ### Author Response · Authors · 2023-11-22
> **Summary of Author Response to All the Reviewers - P2**
>
> **Q2. Manifolds with dimensions of 3 or lower than 3 are too easy and have few applications.**
>
> Understanding the structure of **low-dimensional** manifolds (specifically in 3 and 4 dimensions) present **greater challenges** compared to their high-dimensional counterparts (5 dimensions and above). While 1-dimensional and 2-dimensional manifolds are relatively simpler, the complexity escalates with 3-manifolds. This is exemplified by the famous Poincaré Conjecture for 3-manifolds, which took a century to prove and asserts that every simply connected, closed 3-manifold is homeomorphic to the 3-dimensional sphere.
>
> In four dimensions, the situation is even more intricate. Unlike the three-dimensional case, where Thurston's Geometrization Conjecture (proven by Grigori Perelman) offers a comprehensive framework, four dimensions lack a similar unifying conjecture. This complexity is partly due to the existence of exotic $\mathbb{R}^4$s, which are differentiable manifolds homeomorphic, but not diffeomorphic, to the Euclidean space $\mathbb{R}^4$.
>
> For manifolds of five dimensions and higher, certain aspects of the theory are simplified, thanks in part to the h-cobordism theorem and surgery theory. The h-cobordism theorem, applicable to manifolds of dimension five and above, facilitates understanding their structure by stating that cobordant manifolds under certain conditions are diffeomorphic. However, these tools don't provide a complete classification of high-dimensional manifolds. The problem of determining whether two high-dimensional manifolds are homeomorphic is as complex as the word problem in group theory, which is known to be undecidable.
>
> Given these complexities, studying manifolds of dimension higher than four requires a fundamentally different approach. It is not feasible to leap into high-dimensional cases without thoroughly understanding the lower-dimensional (3D and lower) scenarios. Presently, the literature only focuses on special cases like $\mathbb{R}^d$ balls [1] and smooth regular simple curves [2]. Our work lays the groundwork for future exploration into the high-dimensional realm, recognizing the distinct methodologies required for these advanced studies.
>
> We also appreciate the reviewers' insights on the practical relevance of low-dimensional manifold theory. This area of study is indeed pivotal in many real-world scenarios where high-dimensional data is frequently simplified into lower-dimensional (2 or 3 dimensions) representations for more effective analysis and visualization. Techniques such as Principal Component Analysis (PCA), t-Distributed Stochastic Neighbor Embedding (t-SNE), and Uniform Manifold Approximation and Projection (UMAP) are deeply rooted in the principles of low-dimensional manifold theory. These methods are crucial in transforming complex, high-dimensional data into comprehensible and actionable forms.
>
> Contrary to the view that low-dimensional manifolds may be less relevant, many datasets widely used in machine learning and computer vision intrinsically exhibit 2 or 3 dimensional manifold structures. A prime example is the MNIST dataset, comprising handwritten digits, which can be considered to reside on a manifold with a dimension much lower than its ambient space. This lower-dimensional representation captures the essential features of the data, facilitating more efficient processing and analysis. Similarly, the LineMOD (LM) dataset, which is integral to object recognition and pose estimation tasks, underscores the importance of understanding the underlying low-dimensional manifold. Grasping the structure of these manifolds is not only theoretically insightful but also practically essential for the development of effective algorithms.
>
> In light of these examples, low-dimensional manifold theory emerges not just as a theoretical curiosity but as a fundamental tool in data analysis and algorithm development.

---

> ### Author Response · Authors · 2023-11-22
> **Summary of Author Response to All the Reviewers - P3**
>
> **Q3. The Betti number is not accessible for data as a point cloud.**
>
> In [3], it has been demonstrated that a **manifold's homology** can be **accurately reconstructed** from a sufficiently dense sample of points. Once the homology group is determined, the computation of Betti numbers, which reveal the quantity and types of holes in a manifold, becomes relatively straightforward. Therefore, employing Betti numbers in analytical processes is generally straightforward and effective. However, the primary challenge often lies in acquiring a data sample that is dense enough. Such sufficient sample density is a fundamental requirement in manifold hypothesis studies like those seen in [4], where the results depend critically on manifold properties like curvature or injective radius. These properties, not directly inferable from point cloud data, are often presupposed as known characteristics of the underlying manifold.
>
> This presumption underscores a significant gap in our capacity to fully comprehend manifold structures from sparse datasets, thereby highlighting the necessity for more robust methodologies in manifold learning. These methods must adeptly navigate the common challenge of data limitations. Nevertheless, the theoretical bounds discussed in these studies are far from redundant. They lay down a vital theoretical framework, deepening our understanding of the structures latent in point clouds. This fundamental insight not only drives the development of algorithms but also aids in evaluating the quality of point cloud representations, effectively bridging advanced theoretical concepts with their practical applications. Moreover, these bounds serve as a catalyst for future research, particularly in adapting and applying them more directly to point cloud data.
>
> [1] Itay Safran and Ohad Shamir. Depth-width tradeoffs in approximating natural functions with neural networks. ICML, 2016.
>
> [2] Sam Buchanan, Dar Gilboa and John Wright. Deep Networks and the Multiple Manifold Problem. ICLR 2021.
>
> [3] Partha Niyogi, Stephen Smale, and Shmuel Weinberger. Finding the homology of submanifolds with high confidence from random samples. Discrete & Computational Geometry, 2008.
>
> [4] Hariharan Narayanan and Sanjoy Mitter. Sample complexity of testing the manifold hypothesis. NeurIPS, 2010.

---

### Meta-Review · Area_Chair_Y2yL · 2023-12-06

**Metareview:**

This paper presents an upper bound on the size of ReLU neural networks for the problem of binary classification, when each class is drawn from a smooth low-dimensional manifold. In the setting of what the authors call "solid manifolds" ((intrinsic dimension at most 3, compact, orientable, with boundary), an upper bound is given with two separate components, namely topological complexity - via the Betti numbers, and geometrical complexity - via the condition number. Some synthetic experimental results are presented to demonstrate the correlation between topological complexity and needed network size to sufficiently classify the data.

Reviewers generally agree that the theoretical analysis is interesting. However, there are concerns over the current manuscript:

- The assumption on "solid manifolds" is quite limited and not realistic (Reviewers DRpb, 9Qss, ai47). It needs to restrict to manifolds of very low dimensions (3 or less), compact, with boundary (thus cannot deal with low-dimensional manifolds without boundary such as the 2-dim sphere). The methodology presented here is not generalizable to higher dimensions.

- The experiments presented are too simple and unrealistic to clearly demonstrate the tightness of the bound in terms of the Betti numbers (Reviewer mSxa), as has been claimed.

**Justification For Why Not Higher Score:**

The theoretical methodology is restricted and the experimental validation should be improved.

**Justification For Why Not Lower Score:**

N/A

---

### Decision · Program_Chairs · 2024-01-16

Reject